# Differentially Private Gomory-Hu Trees

**Anders Aamand**
BARC, University of Copenhagen
aa@di.ku.dk

**Justin Y. Chen**
Massachusetts Institute of Technology
justc@mit.edu

**Mina Dalirrooyfard**
Morgan Stanley
minad@mit.edu

**Slobodan Mitrović**
UC Davis
smitrovic@ucdavis.edu

**Yuriy Nevmyvaka**
Morgan Stanley
yuriy.nevmyvaka@morganstanley.com

**Sandeep Silwal**
UW-Madison
silwal@cs.wisc.edu

**Yinzhan Xu**
UC San Diego
xyzhan@ucsd.edu

## Abstract

Given an undirected, weighted $n$-vertex graph $G = (V, E, w)$, a Gomory-Hu tree $T$ is a weighted tree on $V$ that preserves the Min-$s$-$t$-Cut between any pair of vertices $s, t \in V$. Finding cuts in graphs is a key primitive in problems such as bipartite matching, spectral and correlation clustering, and community detection. We design a differentially private (DP) algorithm that computes an approximate Gomory-Hu tree. Our algorithm is $\varepsilon$-DP, runs in polynomial time, and can be used to compute $s$-$t$ cuts that are $\tilde{O}(n/\varepsilon)$-additive approximations of the Min-$s$-$t$-Cuts in $G$ for all distinct $s, t \in V$ with high probability. Our error bound is essentially optimal, since [29] showed that privately outputting a single Min-$s$-$t$-Cut requires $\Omega(n)$ additive error even with $(\varepsilon, \delta)$-DP and allowing for multiplicative error. Prior to our work, the best additive error bounds for approximate all-pairs Min-$s$-$t$-Cuts were $O(n^{3/2}/\varepsilon)$ for $\varepsilon$-DP [47] and $\tilde{O}(\sqrt{mn}/\varepsilon)$ for $(\varepsilon, \delta)$-DP [66], both achieved by DP algorithms that preserve all cuts in the graph. To achieve our result, we develop an $\varepsilon$-DP algorithm for the Minimum Isolating Cuts problem with near-linear error, and introduce a novel privacy composition technique combining elements of both parallel and basic composition to handle 'bounded overlap' computational branches in recursive algorithms, which maybe of independent interest.

## 1 Introduction

Over the last two decades, there has been a significant attention to *privatizing* graph algorithms (see [70, 51, 57, 47, 10, 58, 37, 17, 78, 31, 39, 22, 32, 29, 66, 53] and references within). Graph algorithms are often applied to large data sets, such as social networks, containing sensitive information. It is well understood by now that even minor negligence in handling user data can severely impact privacy; see [8, 74, 59, 82, 26] for a few examples. *Differential privacy (DP)*, introduced by Dwork, McSherry, Nissim, and Smith in their seminal work [36], is a widely adopted standard for formalizing the privacy guarantees of algorithms. Informally, an algorithm is differentially private if the outputs for two given *neighboring* inputs are statistically indistinguishable.

A major part of the literature on private graph algorithms has been approximating cuts which is the setting of our paper. Given an undirected, weighted graph $G = (V, E, w)$ with positive edge weights, a cut is a bipartition of vertices $(U, V \setminus U)$, and the value of the cut is the sum of the weights of edges crossing the bipartition. Given a pair of distinct vertices $s, t \in V$, the Min-$s$-$t$-Cut is

a minimum-valued cut $(U, V \setminus U)$ where $s \in U$ and $t \in V \setminus U$. Min-$s$-$t$-Cut is dual to the Max-$s$-$t$-Flow problem, and the celebrated max-flow min-cut theorem states that the value of the Min-$s$-$t$-Cut equals the value of the Max-$s$-$t$-Flow [41, 38]. Finding a Min-$s$-$t$-Cut (or equivalently Max-$s$-$t$-Flow) is a fundamental problem in algorithmic graph theory, which has been studied for over seven decades (earliest references include [30, 50, 41, 38]), and has inspired ample algorithmic research and applications, including edge connectivity [71], bipartite matching (see, e.g., [25]), minimum Steiner cut [62], vertex-connectivity oracles [77], among others (see the survey [23]).

Over the recent years, nearly tight algorithms have been developed on outputting private Min-$s$-$t$-Cut [29], private global Min Cut [46], and private All Cuts problems [47, 9, 37, 66, 67].[1] However, no work has been done on private All-Pairs Min Cut (APMC) which is our focus. Given an input graph, the goal of APMC is to output a Min-$s$-$t$-Cut for all the pairs of vertices $s$ and $t$ in $V$. We fill this gap, and obtain a private algorithm on APMC with the *same* error guarantee as private Min-$s$-$t$-Cut.

The definition of neighboring inputs depends on the specific application and yields semantically different privacy guarantees. We now specify the standard privacy model for graph cut problems which is used in ours and the aforementioned works. The graph's vertex set is publicly available, and two neighboring graphs differ in only one edge. If the graphs are weighted, two neighboring graphs are those whose total weights differ by at most one and in a single edge. Semantically, this protects privacy if the inclusion of any individual affects the weight of an edge by a bounded quantity.[2] Given neighboring inputs $G$ and $G'$, and a subset of outputs $O$, an $(\varepsilon, \delta)$-DP algorithm $A$ satisfies $\mathbb{P}(A(G) \in O) \leq e^{\varepsilon} \, \mathbb{P}(A(G') \in O) + \delta$. When $\delta = 0$, the algorithm satisfies *pure* DP, and otherwise *approximate* DP.[3] There is always a trade-off between privacy and accuracy: algorithms for nontrivial problems satisfying DP must have errors.

Since the above privacy model is standard for analyzing cut problems under DP, it is natural to also adopt it for the APMC problem. To further motivate the study of APMC under this particular privacy model from a more practical perspective, note that the graph could correspond to a road network and the weight of an edge could correspond to the number of people who travel along the road corresponding to that edge. The above model then ensures that whether any particular individual uses any particular road cannot be deduced based on the output of a DP-algorithm run on the graph. Moreover, minimum cuts are a natural measure of bottlenecks in transportation networks, and it may therefore be of interest to compute them on such traffic graphs in a differentially private fashion.

We now discuss the relevant private algorithms for cut problems. Dalirrooyfard, Mitrović, and Nevmyvaka [29] recently gave the optimal $\varepsilon$-DP algorithm for the Min-$s$-$t$-Cut problem with additive error $O(n/\varepsilon)$. They show an essentially matching $\Omega(n)$ lower bound, even for algorithms which satisfy only approximate DP and allow both multiplicative and additive error.

For the problem of *global Min Cut*, where one seeks the cut minimizing Min-$s$-$t$-Cut over all pairs of node $s, t$, Gupta et al. [46] gave an $\varepsilon$-DP algorithm with additive error $O(\frac{\log n}{\varepsilon})$. Their algorithm runs in exponential time, but they also give a version running in polynomial time but only satisfying approximate DP.[4] The authors also show that there does not exist an $\varepsilon$-DP algorithm for global Min Cut incurring less than $\Omega(\log n)$ additive error.

The private *All Cuts* problem, where one seeks to output a synthetic, private graph which preserves the value of all the cuts, has been extensively studied. Given a graph $G$, the goal is to output a synthetic graph $H$ on the same vertices such that each cut-value in $H$ is the same as the corresponding cut-value in $G$, up to some additive error. Gupta, Roth, and Ullman [47] and, independently, Blocki, Blum, Datta, and Sheffet [9] gave algorithms for this problem with additive error of $O\left(n^{1.5}/\varepsilon\right)$ while satisfying pure and approximate DP, respectively. Eliáš, Kapralov, Kulkarni, and Lee [37] improved

---

[1]Note that the focus of all of these works and our paper is on outputting the actual cut structure not the value of the cut.

[2]We note that there are several other notions of neighboring datasets for other graph problems implying different privacy semantics (including allowing a vertex and all of its edges to change or fixing the unweighted topology of the graph and only allowing edge weights to change). The notion which we use on is the standard for cut problems (used in [46, 9, 37, 84, 29, 66]) as it is the strongest form of privacy that allows for any approximation of cuts or cut values. Moreover, this notion carries over to weight-DP, where in the two neighboring graphs all edges can change in total $\ell_1$ distance 1. We include a detailed discussion of various forms of differential privacy on graphs in Appendix A.5.

[3]The parameter $\delta$ corresponds to the small probability that an individual's data is leaked.

[4]In this case the additive error has a dependency on $\delta$, we do not mention this dependency for simplicity.

| Problem | Additive Error | DP | Output | Runtime |
|---------|----------------|-----|--------|---------|
| Global Min Cut [46] | $\Theta(\log(n)/\varepsilon)$ | Pure | Cut | Exponential |
| Global Min Cut [46] | $\Theta(\log(n)/\varepsilon)$ | Approx | Cut | Polynomial |
| Min-$s$-$t$-Cut [29] | $O(n/\varepsilon)$ and $\Omega(n)$ | Pure | Cut | Polynomial |
| All Cuts [47] | $O(n^{3/2}/\varepsilon)$ | Pure | Synthetic Graph | Polynomial |
| All Cuts [37, 66] | $\tilde{O}(\sqrt{mn}/\varepsilon)$ and $\Omega(\sqrt{mn}/\varepsilon)$ | Approx | Synthetic Graph | Polynomial |
| APMC Values (Trivial) | $O(n/\varepsilon)$ | Approx | Values Only | Polynomial |
| **APMC (Our Work)** | $\tilde{O}(n/\varepsilon)$ | Pure | GH-Tree | Polynomial |

Table 1: State-of-the-art bounds for private cut problems. Dependencies on the approx. DP parameter $\delta$ are hidden. The APMC result with approx. DP follows from advanced composition by adding $\mathrm{Lap}(O(n/\varepsilon))$ random noise to all $\binom{n}{2}$ true values. For APMC, the lower bound of $\Omega(n)$ error [29] also applies as APMC generalizes Min-$s$-$t$-Cut.

on these results for sparse, unweighted (or small weight) graphs, achieving error $\tilde{O}\left(\sqrt{\frac{mn}{\varepsilon}}\right)$[5] with approximate DP. The authors show that this error is essentially tight for algorithms with purely additive error (and no multiplicative approximation). In a follow-up work, Liu, Upadhyay, and Zou [66] extended these results to weighted graphs and gave an algorithm for releasing a synthetic graph for the All Cuts problem with $\tilde{O}\left(\frac{\sqrt{mn}}{\varepsilon}\right)$ error. Recently, Liu et al. [67] gave an algorithm for the same problem with worse error $\tilde{O}\left(\frac{m}{\varepsilon}\right)$ but which runs in near-linear time.

One important cut problem *absent in the study of private graph algorithms* is All-Pairs Min Cut (APMC), which has been extensively studied in the graph algorithms community for over six decades. In a seminal paper, Gomory and Hu [44] showed that there is a tree representation for this problem, called a GH-tree or cut tree, which takes only $n-1$ Min-$s$-$t$-Cut (Max Flow) calls to compute. Consequently, there are only $n-1$ different minimum cut values in an arbitrary graph with positive edge weights. There has been a long line of research in designing faster GH-tree algorithms (e.g. [49, 15, 2, 88, 63, 3, 5, 6], also see the survey [75]), culminating in an almost linear time algorithm for computing the GH-tree [6].

Beyond its importance in graph algorithms, the *all-pairs* aspect of APMC is especially important in the context of DP. Answering multiple queries degrades privacy, so a key feature of differential privacy is the ability to control this degradation through composition theorems (see, for instance, [34, 83]). Applying advanced composition of private mechanisms to the $O(n/\varepsilon)$ error result of [29] for a single Min-$s$-$t$-Cut implies that APMC can be solved with $O(n^2/\varepsilon)$ additive error while satisfying approximate DP. Given the structure of the APMC problem as characterized by the existence of GH-trees, it is natural to ask if one can improve upon black-box composition results.[6]

Existing works provide a preliminary answer. Since the algorithm by [66] approximately preserves all cuts, it can also be used to solve APMC with approximate DP and additive error of $\tilde{O}\left(\sqrt{mn}/\varepsilon\right)$. Additionally, the $\Omega(n)$ lower bound for Min-$s$-$t$-Cut of [29] also applies to APMC, as it is a harder problem. So, in contrast to computing global Min Cut [46], Min-$s$-$t$-Cut [29], and All Cuts [37, 66], where the privacy/error tradeoff is tightly characterized up to $\mathrm{poly}(\log n, 1/\varepsilon)$ factors, there remains a gap of $\approx \sqrt{m/n}$ between the best known lower and upper bound for DP APMC, which can be as large as $\Omega(\sqrt{n})$ in dense graphs. This motivates the following question, which is our focus:

> **Question 1.** *Can we obtain tight bounds on the additive error for APMC with DP?*

## 1.1 Our Results

Our main contribution is an $\varepsilon$-DP algorithm for APMC with $\tilde{O}(n/\varepsilon)$ additive error. Our algorithm privately outputs *all* the Min-$s$-$t$-Cuts while incurring the same error, up to $\mathrm{polylog}(n)$ factors, required to output a Min-$s$-$t$-Cut for a *single* pair of vertices $s$ and $t$. To achieve this result, we introduce an algorithm that solves the more general problem of privately generating an approximate Gomory-Hu tree (GH-tree). Gomory and Hu [44] showed that for any undirected graph $G$, there exists a tree $T$ defined on the vertices of graph $G$ such that for all pairs of vertices $s, t$, the Min-$s$-$t$-Cut in $T$ is also a Min-$s$-$t$-Cut in $G$. We develop a private algorithm for constructing such a tree.

**Theorem 1.1.** *Given a weighted graph $G = (V, E, w)$ with positive edge weights and a privacy parameter $\varepsilon > 0$, there exists an $\varepsilon$-DP algorithm that outputs an approximate GH-tree $T$ with additive error $\tilde{O}(n/\varepsilon)$: for any $s \neq t \in V$, the Min-$s$-$t$-Cut on $T$ and the Min-$s$-$t$-Cut on $G$ differ in $\tilde{O}(n/\varepsilon)$ in cut value with respect to edge weights in $G$. The algorithm runs in time $\tilde{O}(n^2)$, and the additive error guarantee holds with high probability.*

Theorem 1.1 essentially outputs a synthetic graph in a DP manner that approximates each Min-$s$-$t$-Cut with an additive error of $\tilde{O}(n/\varepsilon)$. Since the GH-tree output by Theorem 1.1 is private, any post-processing on this tree is also private. This yields the following corollary.

**Corollary 1.1.** *Given a weighted graph $G$ with positive edge weights and a privacy parameter $\varepsilon > 0$, there exists an $\varepsilon$-DP algorithm that outputs, for all the pairs of vertices $s$ and $t$, a cut whose value is within $\tilde{O}(n/\varepsilon)$ from the value of the Min-$s$-$t$-Cut with high probability.*

Another corollary of Theorem 1.1 is a polynomial-time, pure DP algorithm for global Min-Cut.

**Corollary 1.2.** *Given a weighted graph $G$ with positive edge weights and a privacy parameter $\varepsilon > 0$, there exists an $\varepsilon$-DP algorithm that outputs an approximate global Min-Cut of $G$ in $\tilde{O}(n^2)$ time and has additive error $\tilde{O}(n/\varepsilon)$ with high probability.*

Prior work obtained an exponential-time pure DP algorithm and a polynomial-time approximate DP for global Min-Cut with error $O(\log n/\varepsilon)$ [46]. It is an open question whether there exists an *efficient* algorithm which satisfies pure DP and outputs an approximage global Min-Cut with $\mathrm{polylog}(n)/\varepsilon$ additive error.

Lastly, we note another application of Theorem 1.1 is a *pure*-DP algorithm for *min $k$-cut*[7] problem with multiplicative error 2, additive error $\tilde{O}(nk/\varepsilon)$. No prior poly time pure DP algorithm can compute min $k$-cut on weighted graphs with near-linear in $n$ error. See Corollary F.1) for details.

**Tightness of Our Main Result**   Corollary 1.1 is tight up to $\mathrm{polylog}(n)$ and $\frac{1}{\varepsilon}$ factors since any DP algorithm outputting the Min-$s$-$t$-Cut for a *single* fixed pair of vertices $s$ and $t$ requires $\Omega(n)$ additive error [29]. Thus, our result shows that we can compute *all* min-cuts privately with the same error *required for a single cut* up to log factors. We also note that the $\Omega(n)$ lower bound for a single cut of [29] is on *sparse* graphs and applies to both pure and *approx* DP. Hence, there can be no polynomial improvement on our result even if the input is sparse or if we relax to approx DP.

**Paper Organization**   In the rest of the main body, we give a detailed overview of the challenges in privately creating a GH-tree as well as a technical overview of our approach. Due to space constraints, all pseudocode and proofs are given in the appendix. Two technical ingredients which may be generally applicable are (1) an $\varepsilon$-DP, $\tilde{O}(n/\varepsilon)$ additive error algorithm for Minimum Isolating Cuts [62, 3], a recently introduced problem which has found success as a subroutine in fast algorithms for cut problems, and (2) a general theorem for privacy composition on recursive algorithms where sensitive data is not partitioned into disjoint sets on each recursive call (in this setting, composition is straightforward), but rather the recursive subsets have "bounded-overlap."

**Open Problems.**   We highlight three interesting open problems related to our work in Appendix G.

---

[5]In this work, the notation $\tilde{O}(x)$ stands for $O(x \cdot \mathrm{polylog}\, x)$.

[6]A recent line of work on approximating all-pairs shortest path distances with differential privacy has the same goal of using graph structure to limit the error necessary to answer many queries [81, 39, 22, 13].

[7]The goal is to partition the vertex set into $k$ pieces and the cost of a partitioning is the total weight of all edges between different pieces in the partition. We wish to find the smallest cost solution.

## 2 Technical Overview

A greatly simplified view of a typical approach to designing a DP algorithm begins with a non-DP algorithm, which is modified to ensure privacy. The primary challenge lies in finding an appropriate method to privatizing the algorithm, if such a method even exists, and rigorously proving that satisfies DP. For instance, Gupta et al. [46] employ Karger's algorithm [56] to produce a set of cuts and then use the Exponential Mechanism [70] to choose one of those cut. This simple but clever approach results in an $(\varepsilon, \delta)$-DP algorithm for global Min Cut, for $\delta = \frac{1}{\text{poly}(n)}$, with $O(\frac{\log n}{\varepsilon})$ additive error. Dalirrooyfard et al. [29] show that the following simple algorithm yields $\varepsilon$-DP Min-$s$-$t$-Cut with $O(n/\varepsilon)$ additive error: for each vertex $v$, add an edge from $v$ to $s$ and from $v$ to $t$ with their weights chosen from the exponential distribution with parameter $1/\varepsilon$; return the Min-$s$-$t$-Cut on the modified graph. In the remainder of this section, we first explain why directly privatizing certain existing non-private algorithms fails to achieve the desired additive error. After, we describe our approach.

### 2.1 Obstacles in Privatizing the Algorithm of Gomory and Hu

The All-Pairs Min-Cut (APMC) problem produces cuts for $\binom{n}{2}$ pairs of vertices; it is known that these cuts have only $O(n)$ *distinct* values. This property was leveraged in the pioneering work by Gomory and Hu [44], who introduced the Gomory-Hu tree (GH-tree), a structure that succinctly represents all the Min-$s$-$t$-Cuts in a graph.

The original GH-tree algorithm uses a recursive construction, solving the Min-$s$-$t$-Cut problem at each recursion step: In each step of the recursive call, the input will be a graph $H$ and a special set of terminal vertices $R \subseteq V$. (1) The root of the recursion starts with $H = G$ and $R = V$. (2) At each recursive step, select two arbitrary vertices $s$ and $t$ within $R$ (if $|R| = 1$, then the problem becomes trivial and the recursion stops). (3) Compute the Min-$s$-$t$-Cut in the graph, and say the Min-$s$-$t$-Cut is $(U, V \setminus U)$ where $s \in U, t \notin U$. (4) Create two graphs, $H_s$ and $H_t$, where $H_s$ is the graph with $V \setminus U$ contracted, and $H_t$ is the graph with $U$ contracted. Then recursively solve the problem on $H_s$ with terminal set $R \cap U$, and on $H_t$ with terminal set $R \cap (V \setminus U)$. (5) Finally, we combine the two trees created by the two recursive calls, argue the correctness utilizing the *submodularity* of cuts. The GH-tree efficiently represents all pairwise Min-$s$-$t$-Cuts in the graph by iterating through these steps until there is no supernode of size larger than 1.

To privatize this algorithm, the Min-$s$-$t$-Cut procedures can be replaced with the private Min-$s$-$t$-Cut algorithm introduced in [29]. However, several challenges arise in ensuring low error with this approach. First, the algorithm of [29] modifies the graph. It is unclear whether these modifications should persist in each Min-$s$-$t$-Cut call or if the graph should revert to its original form. Second, the recursion depth may reach $O(n)$. Even ignoring the propagation of error across recursive calls, dependent invocations of the DP Min-$s$-$t$-Cut must use very small values of $\varepsilon$ due to privacy composition. Under basic composition [35], each call to the algorithm of [29] must use $\varepsilon' = O(\varepsilon/n)$ to guarantee $\varepsilon$-DP for the final algorithm. Even using advanced composition [34] would require $\varepsilon' = O(\varepsilon/\sqrt{n \log(1/\delta)})$ to achieve $(\varepsilon, \delta)$-DP. The resulting error would still be higher than that achieved in the prior work preserving all cuts. Finally, we may hope that a single edge impacts only a small number of min-cuts, a property that could be leveraged in constructing a private GH-tree. However, as we depict in Figure 3, the $\ell_1$-sensitivity of changing a single edge is $\Omega(n)$, even when outputting only the values of the Min-$s$-$t$-Cuts.

### 2.2 Towards Privatizing a Low-Depth Algorithm

The preceding discussion indicates that low recursive depth is a crucial property of a private algorithm for producing a GH-tree. While the canonical algorithm of [44] has a linear recursive depth, recent breakthroughs in fast GH-tree algorithms offer the additional advantage of polylogarithmic depth, e.g., [3, 61, 4, 64, 6]. At a high-level, our result derives from privatizing the algorithm described in [61, Section 4.5] (this same algorithm also appears in [4]). To replace components of this algorithm with differentially private counterparts necessitates adding noise, which introduces additive errors throughout the recursive algorithm. Tracking the propagation of this error throughout the algorithm requires careful accounting. Moreover, the specific steps of the non-private algorithm presents several fundamental challenges in creating a private version. The remainder of this subsection briefly summarizes the structure of this algorithm.

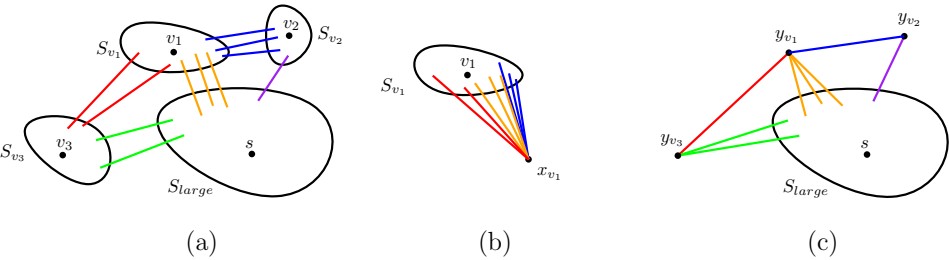

Figure 1: Recursive structure of our low-depth GH-tree. (a) The decomposition into $(S_v)_{v \in R}$ and $S_{large} = V \setminus \bigcup_{v \in R} S_v$ defines the recursive subinstances. For each $v \in R$, $S_v$ constitutes a Min-$s$-$v$-Cut. None of the $S_v$ or $S_{large}$ are too large, which leads to a polylogarithmic upper-bound on the recursion depth of the algorithm. (b) A subinstance obtained by contracting $V \setminus S_{v_1}$ to a single vertex $x_{v_1}$. (c) A subinstance obtained by contracting $S_v$ to a single vertex $y_v$ for each $v \in B$.

**Minimum Isolating Cuts.** For a set $U$ of vertices in $G$, let $w(U)$ be the sum of the weights of the edges in $G$ with exactly one endpoint in $U$. The first building block is a subroutine called *Minimum Isolating Cuts*, introduced in [62, 3].

**Definition 2.1** (Min Isolating Cuts [62, 3])**.** *Given a set of terminals $R \subseteq V$, the Min Isolating Cuts problem asks to output a collection of sets $\{S_v \subseteq V : v \in R\}$ such that for each vertex $v \in R$, the set $S_v$ satisfies $S_v \cap R = \{v\}$, and it has the minimum value of $w(S_v')$ over all sets $S_v'$ that $S_v' \cap R = \{v\}$. In other words, each $S_v$ is the minimum cut separating $v$ from $R \setminus \{v\}$.*

The Isolating Cuts Lemma [62, 3] shows how to solve Min Isolating Cuts in $O(\log n)$ calls to Min-$s$-$t$-Cut. At a high level, the Isolating Cuts Lemma is used to simultaneously find disjoint Min-$s$-$t$-Cuts for several $s, t$ pairs. More precisely, a single recursive step of the GH-tree algorithm of [61] uses Min Isolating Cuts to find a collection of disjoint subsets of vertices $\{S_v \subset V\}_{v \in R}$ for some set of terminals $R \subset V$ and a source vertex $s \notin R$ which satisfy three properties:

(a) The subsets correspond to Min-$s$-$v$-Cuts: each subset $S_v$ is the $v$-side of a Min-$s$-$v$-Cut.
(b) The subsets are not too large: $|S_v| \leq (1 - \Omega(1))n$.
(c) The union of the subsets is not too small: $\sum |S_v| \geq n/\operatorname{polylog}(n)$.

The algorithm is recursively applied to each of the subsets $S_v$ (with vertices outside of $S_v$ contracted) as well as the remainder of the vertices $S_{large} = V \setminus \bigcup S_v$ (with each $S_v$ contracted). In concert, conditions (b) and (c) guarantee that the recursive depth will be polylogarithmic. Condition (a) allows us to stitch the outcome of these recursive calls together into a GH-tree that preserves Min-$s$-$t$-Cuts. See Figure 1 for a visualization of this process. It remains to describe how to generate the source $s$, terminals $R$, and subsets $\{S_v\}_{v \in R}$ to complete the high level description of the algorithm of [61]. We call a subset $S_v$ having properties (a) and (b) a *good* subset.

**Choosing $s$.** The vertex $s$ is chosen uniformly due to the following: Let $D^*$ be the set of all vertices $v \in V \setminus \{s\}$ for which there exists a Min-$s$-$v$-Cut with the $v$-side having cardinality at most $n/2$, i.e., the $v$ side of the cut satisfies condition (b). A lemma from [2] shows that, as long as $s$ is chosen uniformly at random from $V$, we have $\mathbb{E}\left[|D^*|\right] = \Omega(n)$: a constant fraction of vertices have small cardinality Min-$s$-$v$-Cuts. This is important since the algorithm chooses $R$ from $D^*$, i.e., all the subsets $S_v$ chosen by the algorithm have cardinality at most $n/2$ and hence they satisfy property (b).

**Finding good subsets $\{S_v\}$ which cover $D^*$.** After choosing $s$, the algorithm selects geometrically decreasing sets of potential terminals $R^0, \ldots, R^{\lfloor \lg n \rfloor}$ where each vertex other than $s$ is selected into $R^i$ uniformly at random with probability $2^{-i}$. For each set of terminals $R^i$, the algorithm runs Min Isolating Cuts on $R^i \cup \{s\}$ and keeps the "good" subsets $\{S_v\}_{v \in R^i}$ output by Min Isolating Cuts. Recall that a good subset $S_v$ has properties (a) and (b). To test property (a), we verify which subsets

$S_v$ correspond to Min-$s$-$v$-Cuts by computing the single source Min-$s$-$v$-Cut values from $s$ to all $v \in V \setminus \{s\}$ and comparing that to the value of the cuts $S_v$. To satisfy property (b), we only keep $S_v$ if $|S_v| \leq n/2$.

The first observation is that for any particular Min-$s$-$v$-Cut with $v$-side $S_v^*$, there is some sampling level $i^*$ where there is a reasonable probability that the only terminal sampled in $S_v^*$ is $v$, i.e. $R^{i^*} \cap S_v^* = \{v\}$. If this event occurs, then Min Isolating Cuts with terminals $R^{i^*} \cup \{s\}$ will return $S_v^*$ as one of its outputs–the minimum cut separating $v$ from the other terminals is the Min-$s$-$v$-Cut. The second observation is that there is some sampling level $i \leq \lfloor \lg n \rfloor$ where, when run on the corresponding terminals $R^i \cup \{s\}$, the expected size of the union of good subsets is $\Omega\left(\frac{|D^*|}{\log^2 n}\right) = \Omega\left(\frac{n}{\log^2 n}\right)$, i.e., satisfying (c). Thus, selecting the sampling level whose good subsets have maximum total cardinality produces a collection $\{S_v\}$ satisfying conditions (a), (b), (c).[8]

## 2.3 Three Core Challenges

As the Min Isolating Cuts are a major building block of the algorithm of [61], our first challenge is finding a privatized version of it.

> **Challenge 1.** *Construct a differentially private, approximate Min Isolating Cuts algorithm.*

Next, in one recursive step, the non-private algorithm compares the Min Isolating Cuts to single source Min-$s$-$v$-Cut values. Making these procedures private will introduce additive errors, and this makes it difficult to satisfy conditions (b) and (c). Recall that in [61], at some sampling level Min Isolating Cuts will correspond to Min-$s$-$v$-Cuts which are neither too large for condition (b) nor too small for condition (c). This convenient property is deduced from the cardinality of the *true* Min-$s$-$v$-Cut and the fact that calls to Min Isolating Cuts will find exactly those Min-$s$-$v$-Cuts. Without adjustment, plugging in additive approximate subroutines will fail to produce recursive sub-instances of a reasonable size, undermining the goal of low recursive depth.

> **Challenge 2.** *Design a recursive step that employs additive approximations for Min-$s$-$v$-Cut values and Min Isolating Cuts while ensuring that (1) no sub-instance is too large and (2) the union of sub-instances is sufficiently large.*

Even assuming that the above two challenges are resolved, it remains unclear how to account for the privacy loss of the final low-depth recursive algorithm. The difficulty lies in the fact that, despite the polylogarithmic recursion depth, a single edge of the original graph may appear in multiple sub-instances across a single recursive level. A given instance with terminal set $R$ will have $|R| + 1$ recursive sub-instances: $|R|$ of them are obtained by contracting each of the vertex sets $(V \setminus S_v)_{v \in R}$ into a single vertex $x_v$, and one is obtained by contracting each $S_v$, $v \in R$ into a single vertex $y_v$ (see Figure 1 (b)). If an edge has both of its incident vertices lying in a single $S_v$ or in $S_{large} := V \setminus \bigcup_{v \in R} S_v$, then that edge will only appear in the corresponding sub-instance. However, an edge with, say, one endpoint in $S_u$ and the other in $S_v$ for different $u$ and $v$ will appear in three recursive sub-instances: those obtained by contracting $V \setminus S_u$, $V \setminus S_v$, and $V \setminus S_{large}$. Moreover, this edge could appear in up to $O(n)$ sub-instances. So, if we naïvely apply basic composition, we need the computation on each sub-instance to be $(\varepsilon/n)$-DP. This would lead to a final $\tilde{O}(n^2/\varepsilon)$ additive error guarantee, which is again worse than previous work on preserving all cuts.

> **Challenge 3.** *Modify the algorithm so that the privacy loss of a given edge can be accounted for by the recursion depth.*

In the following sections, we give a technical overview on how we overcome these challenges.

## 2.4 Addressing Challenge 1: Privatizing Min Isolating Cuts (Appendix B)

[62] and [3] independently introduce the Isolating Cuts Lemma, showing how to solve the Min Isolating Cuts problem using $O(\log |R|)$ many Min-$s$-$t$-Cut calls. [62] use it to find the global Min Cut in polylogarithmically many invocations of Max Flow and [3] use it to compute GH-tree in

---

[8]Note that the final set of terminals $R$ is inferred from the selection of $\{S_v\}$, corresponding to the good subsets at a certain sampling level.

unweighted graphs. Subsequent algorithms for GH-tree also use the Isolating Cuts Lemma, including the almost linear time algorithm for weighted graphs [6]. The Isolating Cuts Lemma has been extended to obtain new algorithms for finding the non-trivial minimizer of a symmetric submodular function and solving the hypergraph minimum cut problem [72, 21]. We develop the first differentially private algorithm for finding Min Isolating Cuts.

**Theorem 2.1.** *There is an $\varepsilon$-DP algorithm that given a graph $G$ and a set of terminals $R$, outputs sets $\{S_v : v \in R\}$, such that for each vertex $v \in R$, the set $S_v$ satisfies $S_v \cap R = \{v\}$, and $w(S_v) \leq w(S_v^*) + \tilde{O}(\frac{n}{\varepsilon})$ with high prob., where $\{S_v^* : v \in R\}$ are the Min Isolating Cuts for $R$.*

Tackling this first challenge—obtaining a DP algorithm for constructing Min Isolating Cuts—turns out to be not too difficult. We repeatedly invoke the private Min-$s$-$t$-Cut algorithm of [29] $O(\log n)$ times. To establish the error bound, we carefully apply the cut submodularity property (Lemma A.1) to account for the approximate errors. Since we are separating a potentially large set of terminals rather than a single pair, we must ensure the *total* error, summed across all terminals, remains bounded by $\tilde{O}\left(\frac{n}{\varepsilon}\right)$ (see Lemma B.1 for this stronger result).

## 2.5 Addressing Challenge 2: Privatizing the Recursive Step (Appendix B and Appendix C)

The second challenge involves controlling the size of the sets produced by our DP Min Isolating Cuts algorithm. To address this, we use the following idea to bias the algorithm toward selecting smaller cardinality sets to satisfy that property (b). Consider a graph $G$ with terminals $s$ and $t$. We aim to find an approximate Min-$s$-$t$-Cut where the $s$-side of the cut contains a small number of vertices, without significantly sacrificing accuracy. To achieve this, we can add edges from $t$ to every vertex with a certain weight, penalizing the placement of vertices on the $s$-side of the cut. Applying this idea to our DP Min Isolating Cuts algorithm, we enforce that if a true minimum isolating cut of size at most $n/2$ exists, we will output an isolating cut of size at most $0.9n$ while increasing the additive error of the algorithm by only a constant factor.

The above only takes care of property (b). A subtlety in the argument of [61] in making sure property (c) is satisfied over all good subsets is the following: Randomly sampling terminals $R^i$ will, with reasonable probability, mean that for some vertices $v$, its Min Isolating Cut $S_v$ will be the same as its Min-$s$-$v$-Cut $S_v^*$. In particular, this will be true if $v \in R^i$ is the only vertex sampled on its side of the Min-$s$-$v$-Cut. In this case, $S_v$ is a good subset and is kept by the algorithm. Later, using that $S_v = S_v^*$, they argue that the union of good subsets is not too small.

Unfortunately, even though the Min Isolating Cuts output by our DP algorithm have small additive error, they can have significantly fewer nodes than the optimal cut $S_v^*$. Essentially, although the true Min-$s$-$v$-Cut may be large, there is no lower bound on the node size of approximate Min-$s$-$v$-Cuts. This poses a challenge in showing that the union of the good sets $S_v$ retained by the algorithm is not too small (property (c)). To address this, we compare $S_v$ not to $S_v^*$, but instead to the *smallest cardinality* "approximate" Min-$s$-$v$-Cut $\tilde{S}_v$. The argument that, for some sampling level, there will be a reasonable probability that Min Isolating Cuts will return an $S_v$ which is an approximate Min-$s$-$v$-Cut and has cardinality lower bounded by $|\tilde{S}_v|$ then goes through. For the complete argument, we adjust our notion of "approximation" based on the size of $\tilde{S}_v$, allowing for weaker approximations for smaller cardinality sets. This only degrades our approximation with respect to property (a) by log factors.

## 2.6 Addressing Challenge 3: Bounding Privacy along the Recursion Tree (Appendix D)

The third challenge involves controlling the privacy budget. We need to ensure that, for any two neighboring graphs, the output distributions across $\text{polylog}(n)$ recursive layers differ by, at most a $e^\varepsilon$ factor. To achieve this guarantee, we allocate the privacy budget across $\text{polylog}(n)$ recursive sub-instances. Recall that a recursion depth of $\text{polylog}(n)$ does not imply that the privacy budget can be evenly allocated across $\text{polylog}(n)$ instances. To recall the 3rd challenge, consider two neighboring instances, $G$ and $G'$, that differ on edge $xy$. Suppose that in the first step of the algorithm, the good sets $\{S_v\}$ and $S_{\text{large}}$ are the same in both graphs $G$ and $G'$. Suppose that $x \in S_v$ and $y \in S_u$. In this case, the edge $xy$ affects multiple recursive instances: specifically, in $G_v$, $G_u$, and $G_{\text{large}}$ (similarly $G_v'$, $G_u'$, and $G_{\text{large}}'$). Thus computations across multiple branches of the recursion depend on the same edge, meaning the privacy guarantee depends on more than just the recursion depth.

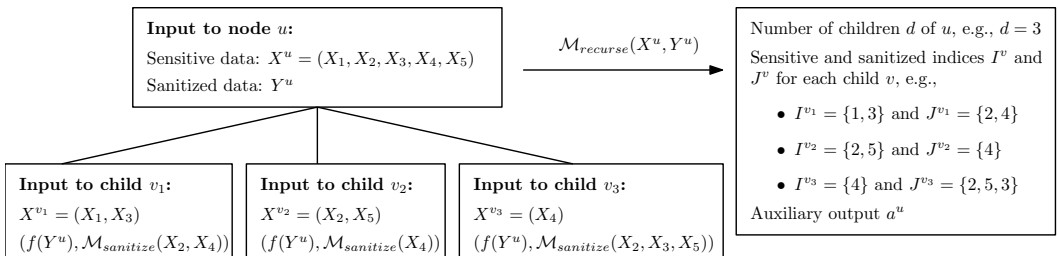

Figure 2: Bounded-overlap branching composition. Node $u$ receives as input a set of sensitive data $X^u$ and a set of sanitized data $Y^u$. Via a DP mechanism $\mathcal{M}_{recurse}$, it then computes the number of children of $u$, and for each child $v$, a set of sensitive indices $I_v$ and a set of sanitized indices $J_v$. Importantly, the sets $I_v$ are disjoint, and an index $j$ can appear in at most $\ell$ different sets $J^v$ (in the figure $\ell = 2$). Node $u$ may further return some other DP output $a^u$. Each child $v$ now receives as input the data $(X_i)_{i \in I^v}$ and the concatenation $(f(Y), \mathcal{M}_{sanitize}(X_j)_{j \in J^v})$ where $\mathcal{M}_{sanitize}$ is a DP mechanism and $f$ is some arbitrary function. Intuitively, if an element $X_i$ of the dataset is sanitized in a child $v$, then post-processing ensures that the entire computation in the subtree rooted at $v$ is DP with respect to user $i$. Since user $i$'s data $X_i$ can only appear unsanitized in a single child, and is sanitized in at most $\ell$ children, we can apply basic composition over $h$ computations of $\mathcal{M}_{recurse}$ and $\ell \cdot h$ computations of $\mathcal{M}_{sanitize}$, where $h$ is the depth of the tree.

To address this issue, we introduce an additional step before recursing on certain new instances. For the instances where all vertices in $V \setminus S_v$ are contracted to a single vertex, we first add an edge from the contracted vertex to every vertex in $S_v$ with weights drawn from $\mathrm{Lap}(\mathrm{polylog}(n)/\varepsilon)$. Then we recurse on this altered graph. The intuition is that in neighboring instances, if $x \in S_v$ and $y \notin S_v$, these noisy edges cancel the effect of $xy$, preventing its influence from propagating further along this branch of the recursion. As a result, $xy$ impacts only $G_{large}$. Fortunately, the added noise only contributes $\tilde{O}(n/\varepsilon)$ to the final additive error, as noise is added to only $O(n)$ edges.

**Bounded-Overlap Branching Composition.**   The approach outlined above is an example of a more general privacy composition technique, which we develop and formalize (Appendix D.2). Here, we provide a high-level overview of the technique and refer to Figure 2 for an illustration. Consider an algorithm $\mathcal{M}_{branch}$ that takes a sensitive dataset $X$ as input, performs DP computation on $X$, and recurses on some subsets of $X$. Now, consider the computation tree corresponding to the recursive branching of $\mathcal{M}_{branch}$. If the subsets assigned to a node's children are chosen privately and the subsets are *disjoint*, privacy composition is straightforward, as outlined below.

Let $h$ upper-bound the depth of the tree. If, at a single node, the release of the computation on $X$ as well as the indices to the subsets of its children is $(\varepsilon, \delta)$-DP, then the entire mechanism $\mathcal{M}_{branch}(X)$ is $(h\varepsilon, \delta h)$-DP. To see this, consider any pair of neighboring datasets $X, X'$ which differ only on the coordinate $i^*$. As the subsets of indices assigned by each node to its children are disjoint, there is a unique path from the root of the tree along which nodes do computation on $X_{i^*}$. All other nodes in the tree do not depend on the value of $X_{i^*}$ after conditioning on the release of the index sets of all children of nodes along the path. Composition along this path gives the result. This can be thought of as a novel combination of parallel[9] and basic composition, and post-processing.

The challenge in our setting is that each node in the recursion tree does *not* partition its data among its children (the same edge weight can appear in multiple subinstances). A single data point may be sent to multiple children, so the number of nodes in the tree whose input includes $X_{i^*}$ may be exponential in the depth $h$. We define a restricted class of recursive mechanisms, which take in both a sensitive dataset $X$ and a sanitized dataset $Y$. The overall recursive mechanism $\mathcal{M}_{branch}$ is formed using two private subroutines $\mathcal{M}_{recurse}$ and $\mathcal{M}_{sanitize}$. At each recursive step, $\mathcal{M}_{recurse}(X, Y)$ is used to generate some partial output, the number of recursive children, and a set of sensitive and sanitized index sets $I = (i_1, \ldots, i_n)$ and $J = (j_1, \ldots, j_m)$ for each recursive child. A child receives as input (1) a sensitive dataset $X_{i_1}, \ldots, X_{i_n}$ which is a subset of its parent's sensitive dataset $X$, and (2) a sanitized input, which is the concatenation of any function of its parent's sanitized dataset $Y$ and the output of the private computation $\mathcal{M}_{sanitize}(X_{j_1}, \ldots, X_{j_m})$.

---

[9]We use parallel composition to refer to the fact that the union of outputs of an $(\varepsilon, \delta)$-DP mechanism applied separately to disjoint subsets of a sensitive dataset is itself $(\varepsilon, \delta)$-DP.

The key property of the mechanisms we define is **bounded-overlap**, parameterized by some constant $\ell$. For any index $i$ into the sensitive dataset $X$ of the parent: (1): There is at most one child whose *sensitive* index set contains $i$. (2): There are at most $\ell$ children whose *sanitized* index sets contain $i$.

**Theorem 2.2** (Informal Version of Theorem D.2). *Let $\mathcal{M}_{branch}$ be a recursive mechanism as described above with bounded-overlap and maximum depth $h$ with subroutines $\mathcal{M}_{recurse}$ and $\mathcal{M}_{sanitize}$ which are $(\varepsilon_1, \delta_1)$-DP and $(\varepsilon_2, \delta_2)$-DP, respectively. Then, releasing the union of outputs of $\mathcal{M}_{recurse}$ and $\mathcal{M}_{sanitize}$ over the entire recursion tree is $(h\varepsilon_1 + (h-1)\ell\varepsilon_2, h\delta_1 + (h-1)\ell\delta_2)$-DP.*

Note that this generalizes the above example, where all children get disjoint subsets of the parent's data by setting $\ell = 0$. Indeed, the high-level idea of the proof of this theorem follows the same argument, with more care taken to argue that, by sanitizing the overlapping data sent to a node's children, we can bound privacy along a single path from the root of the tree for any particular index $i^*$. We ultimately prove the privacy of our GH-tree algorithm by the application of this general theorem. As described earlier in this subsection, the sanitization procedure is to add Laplace noise from the contracted node to all other nodes in $S_v$ subinstances. Then, each edge is only part of the private input to a single recursive child. A key observation is that the recursive GH-tree algorithm has bounded overlap with $\ell = 2$ as any given edge can only belong to two such $S_v$ instances.

## Acknowledgements

A. Aamand was supported by the VILLUM Foundation grant 54451. J. Chen was supported by an NSF Graduate Research Fellowship under Grant No. 17453. S. Mitrović was supported by the Google Research Scholar and NSF Faculty Early Career Development Program No. 2340048. Y. Xu was supported by NSF Grant CCF-2330048, HDR TRIPODS Phase II grant 2217058, and a Simons Investigator Award. Part of this work was conducted while J. Chen and S. Mitrović were visiting the Simons Institute for the Theory of Computing.

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

# A  Preliminaries

## A.1  Notation

We use $G = (V, E, w)$ to denote a weighted, undirected graph with vertex set $V$, edge set $E$ and edge weights $w$. For a subset of vertices $S \subseteq V$, we use $\partial_G S$, or simply $\partial S$ when $G$ is clear from context, to denote the set of edges between $S$ and $V \setminus S$. For a set of edges $Q \subseteq E$, we use $w(Q)$ to denote the sum of the weights of the edges in $Q$. For a set of vertices $S \subseteq V$, we use $w(S)$ to denote $w(\partial S)$, for simplicity.

For $s, t \in V$, we use $\lambda_G(s, t)$ to denote the Min-$s$-$t$-Cut value in a specified graph $G$. When clear from context, the subscript $G$ might be omitted. Throughout our computation, we assume that all Min-$s$-$t$-Cuts or Min Isolating Cuts are unique. This is without loss of generality by adding small noise to the edges and applying the isolation lemma [73] (see also [16, 2]). [10] For $n > 0$, $\lg(n)$ is logarithm base-2 and $\ln(n)$ is logarithm base-$e$. Unless specified, high probability refers to failure probability of at most $1/n^{O(1)}$ where we can pick the $O(1)$ factor to be an arbitrarily large constant of our choice.

| Notation | Meaning | Reference |
|---|---|---|
| $\varepsilon$ | privacy parameter | Definition A.4 |
| $\partial_G S, S \subseteq V$ | set of edges between $S$ and $V \setminus S$ in $G$ | Appendix A.1 |
| $w_G(Q), Q \subseteq E$ | sum of weights of edges in $Q$ in $G$ | Appendix A.1 |
| $w_G(S), S \subseteq V$ | sum of weights of edges in $\partial S$ in $G$ | Appendix A.1 |
| $\lambda_G(s, t)$ | min $s$-$t$ cut value in $G$ | Appendix A.1 |
| $S_u$ | isolating cut for a vertex $u$ | Definition 2.1 |
| $\hat{w}_G(\cdot), \hat{S}_u$ | privatized versions of $w_G(\cdot)$ and $S_u$ | Algorithm 2 |

Table 2: A summary of our notation throughout the paper. When clear from the context, we drop the subscript of the graph $G = (V, E, w)$.

## A.2  Graph Cuts

In our algorithms, we use the notion of *vertex contractions*, which we formally define next.

**Definition A.1** (Vertex contractions). *Let $X \subseteq V$ be a subset of vertices of the graph $G = (V, E, w)$. Contracting the set $X$ into a vertex is done as follows: we add a vertex $x$ to the graph and remove all the vertices in $X$ from the graph. Then for every vertex $v \in V \setminus X$, we add an edge from $x$ to $v$ with weight $\sum_{x' \in X} w(x'v)$. Note that if none of the vertices in $X$ has an edge to $v$, then there is no edge from $x$ to $v$.*

We use the submodularity property of cuts in many of our proofs.

**Lemma A.1** (Submodularity of Cuts [27]). *For any graph $G = (V, E, w)$, and any two subsets $S, T \subseteq V$, it holds*

$$w(S) + w(T) \geq w(S \cup T) + w(S \cap T).$$

Recall the definition of Min Isolating Cuts problem (see Definition 2.1). We use the following simple fact.

**Fact A.1** ([62]). *Given a set of terminals $R \subseteq V$, there always exists a set of minimum isolating cuts $\{S_v : v \in R\}$ such that the cuts are disjoint.*

**Definition A.2** (Gomory-Hu Steiner tree [61]). *Given a graph $G = (V, E, w)$ and a set of terminals $U \subseteq V$, the Gomory-Hu Steiner tree is a weighted tree $T$ on the vertices $U$, together with a function*

---

[10]The isolation lemma involves adding noise to existing edge weights. This noise can be bounded in magnitude to be at most $1/\operatorname{poly}(n)$, so scaling the weights down by a $(1+1/\operatorname{poly}(n))$ factor yields the normal neighboring definition for DP.

$f : V \to U$, *such that: For all* $s, t \in U$, *consider the minimum-weight edge* $uv$ *on the unique* $s$-$t$ *path in* $T$. *Let* $U_0$ *be the vertices of the connected component of* $T - uv$ *containing* $s$. *Then, the set* $f^{-1}(U_0) \subseteq V$ *is the Min-s-t-Cut, and its value is* $w_T(uv)$.

*Note that for* $U = V$ *and* $f(v) = v$ *the Gomory-Hu Steiner tree equals the Gomory-Hu tree.*

### A.3 Concentration Inequalities

**Theorem A.1** (Sums of Exponential Random Variables ([54, Theorem 5.1]))**.** *Let* $X_1, \ldots, X_N$ *be independent random variables with* $X_i \sim Exp(a_i)$. *Let* $\mu = \sum_{i=1}^{N} \frac{1}{a_i}$ *be the expectation of the sum of the* $X_i$'s *and let* $a^* = \min_i a_i$. *Then, for any* $t \geq 1$,

$$\mathbb{P}\left[\sum_{i=1}^{N} X_i \geq t\mu\right] \leq \frac{1}{t} \exp[-a^* \cdot \mu(t - 1 - \ln t)]$$

**Corollary A.1.** *Let* $X_1, \ldots, X_N$ *be independent random variables with* $X_i \sim Exp(a)$ *for some real number* $a > 0$. *Let* $\mu = N/a$ *be the expectation of the sum of the* $X_i$'s. *Then, for any* $T \geq 2\mu$,

$$\mathbb{P}\left[\sum_{i=1}^{N} X_i \geq T\right] \leq \exp(-aT/10).$$

*Proof.* Apply Theorem A.1 with $a^* = a$ and $t = T/\mu$, we obtain

$$\begin{aligned}
\mathbb{P}\left[\sum_{i=1}^{N} X_i \geq T\right] &\leq \frac{1}{t} \exp[-a \cdot \mu(t - 1 - \ln t)] \\
&= \exp[-N(t - 1 - \ln t + (\ln t)/N)] \\
&\leq \exp(-Nt/10) \qquad\qquad\qquad\qquad \text{(as } t \geq 2) \\
&= \exp(-aT/10).
\end{aligned}$$

$\square$

### A.4 Differential Privacy

In this paper, we focus on a weighted version of edge differential privacy. At the end of this section, we discuss why this choice of neighboring graphs is made as well as some connections between notions of neighboring graphs.

**Definition A.3** (Edge-Neighboring Graphs)**.** *Graphs* $G = (V, E, w)$ *and* $G' = (V, E', w')$ *are called edge-neighboring if there is* $uv \in V^2$ *such that* $|w_G(uv) - w_{G'}(uv)| \leq 1$ *and for all* $u'v' \neq uv$, $u'v' \in V^2$, *we have* $w_G(u'v') = w_{G'}(u'v')$. *Note that* $w(u''v'') = 0$ *for all non-edges* $u''v''$.

**Definition A.4** (Differential Privacy [36])**.** *A (randomized) algorithm* $\mathcal{A}$ *is* $(\varepsilon, \delta)$-*private (or* $(\varepsilon, \delta)$-*DP) if for any neighboring graphs* $G$ *and* $G'$ *and any set of outcomes* $O \subset Range(\mathcal{A})$ *it holds*

$$\mathbb{P}\left[\mathcal{A}(G) \in O\right] \leq e^{\varepsilon} \, \mathbb{P}\left[\mathcal{A}(G') \in O\right] + \delta.$$

*When* $\delta = 0$, *algorithm* $\mathcal{A}$ *is* pure differentially private, *or* $\varepsilon$-*DP.*

We now state some standard properties of differential privacy which we will utilize in our algorithm design and analysis.

**Theorem A.2** (Basic composition [36, 33])**.** *Let* $\varepsilon_1, \ldots, \varepsilon_t > 0$ *and* $\delta_1, \ldots, \delta_t \geq 0$. *If we run* $t$ *(possibly adaptive) algorithms where the* $i$-th *algorithm is* $(\varepsilon_i, \delta_i)$-*DP, then the entire algorithm is* $(\varepsilon_1 + \ldots + \varepsilon_t, \delta_1 + \ldots + \delta_t)$-*DP.*

**Theorem A.3** (Laplace mechanism [34])**.** *Consider any function* $f$ *which maps graphs* $G$ *to* $\mathbb{R}^d$ *with the property that for any two neighboring graphs* $G, G'$, $\|f(G) - f(G')\|_1 \leq \Delta$. *Then, releasing*

$$f(G) + (X_1, \ldots, X_d)$$

*where each* $X_i$ *is i.i.d. with* $X_i \sim Lap(\Delta/\varepsilon)$ *satisfies* $\varepsilon$-*DP.*

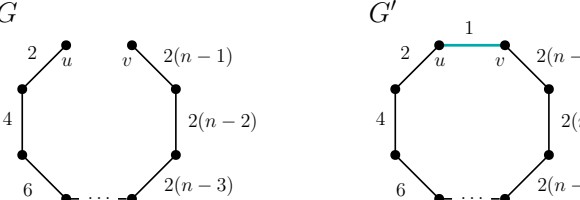

Figure 3: This example describes two weighted graphs, $G$ and $G'$, that differ by a single edge $\{u, v\}$: $G$ is a path, while $G'$ is a cycle. The numbers next to the edges are their weights. Note that both graphs have $n-1$ distinct min-cut values. However, the min-cut values in $G$ are $\{2, 4, \ldots, 2(n-1)\}$, while the min-cut values in $G'$ are $\{3, 5, \ldots, 2(n-1)+1\}$. Thus, the min-cut value sets of these two neighboring graphs differ in $n-1$ entries.

Note that any $w(S)$ for $S \subset V$ has sensitivity $\Delta \leq 1$ as changing one edge weight by one can change the sum of a subset of edge weights by at most one. We now state a result on privately releasing an approximate Min-$s$-$t$-Cut for a single pair of vertices $s, t$.

**Theorem A.4** (Private Min-$s$-$t$-Cut [29])**.** *Fix any $\varepsilon > 0$. There is an $(\varepsilon, 0)$-DP algorithm PrivateMin-s-t-Cut$(G = (V, E, w), s, t, \varepsilon)$ for $s \neq t \in V$ that reports an s-t cut for n-vertex weighted graphs that is within $O(\frac{n}{\varepsilon})$ additive error from the Min-s-t-Cut with high probability.*

By standard techniques, we can also use Theorem A.4 to design an $(\varepsilon, 0)$-DP algorithm for computing an approximate Min-$S$-$T$-Cut for two disjoint subsets $S, T \subseteq V$ that is within $O(\frac{n}{\varepsilon})$ additive error from the actual Min-$S$-$T$-Cut (e.g., by contracting all vertices in $S$ and all vertices in $T$ to two supernodes). Furthermore, our final algorithm is recursive with many calls to Private Min-$S$-$T$-Cut for graphs with few vertices, and it is not enough to succeed with high probability with respect to $n$. The error analysis of [29] shows that the error is bounded by the sum $O(n)$ random variables distributed as $\text{Exp}(\varepsilon)$. Using Corollary A.1 yields the following corollary:

**Corollary A.2** (Private Min-$S$-$T$-Cut)**.** *Fix any $\varepsilon > 0$, there exists an $(\varepsilon, 0)$-DP algorithm PrivateMin-S-T-Cut$(G = (V, E, w), S, T, \varepsilon, \beta)$ for disjoint $S, T \subseteq V$ that reports a set $C \subseteq V$ where $S \subseteq C$ and $C \cap T = \emptyset$, and $w(\partial C)$ is within $O\left(\frac{n + \log(1/\beta)}{\varepsilon}\right)$ additive error from the true min-S-T-cut with probability at least $1 - \beta$.*

## A.5 On various notions of neighboring graphs

For graph data, there are several choices of neighboring datasets with very different semantics for the privacy they correspond to.

At one extreme are vertex-neighboring graphs, where neighboring graphs differ arbitrarily in the edges incident on a single vertex, e.g., [14]. Semantically, each vertex corresponds to a person, and vertex differential privacy protects the data of that individual person. While this offers broad protection, this notion of privacy is simply too restrictive for cut problems. It has been found to be useful in simpler problems, such as estimating the edge density of random graphs. However, the value of any cut can change arbitrarily between neighboring graphs, so no approximation of any cut value is possible while maintaining privacy.

We consider edge-neighboring graphs, which is the standard for cut problems in the literature, see, e.g., [46, 47, 37, 66, 29]. It is the strongest form of privacy for which we can get a meaningful approximation to cut problems. For unweighted graphs, neighboring graphs are those in which a single edge has been added/removed. For weighted graphs, two related notions are considered: where a single edge can change in weight by 1 and where all edges can change in total $\ell_1$ distance 1[11]. Semantically, these notions of privacy are meaningful if individuals impact the existence of edges and the size of edge weights. We use the former definition but note that, as is often the case, our result applies in both settings. We outline the reduction below.

---

[11]Note that both of these capture the unweighted case as zero weight edges are equivalent to edges not belonging to the graph in the context of cut problems.

Let $\mathcal{A}$ be an algorithm satisfying the former notion of edge differential privacy (changing a single edge by weight 1) with an approximation error that depends linearly on $1/\varepsilon$ and is scale-invariant in that it does not explicitly depend on the scale of the edge weights in the graph. Say two graphs have edge weight vectors $w_G, w_{G'} \in \mathbb{R}^{\binom{n}{2}}$ with $\|w_G - w_{G'}\|_1 \le 1$. Let $\Delta = w_G - w_{G'}$, and let $C$ be some constant such that $C\Delta$ is integer-valued – we assume that such a constant exists either due to finite precision of the weights or by rounding the weights to finite precision with arbitrarily small loss. Note that $\|C\Delta\|_1 \le C$. By group privacy, running $\mathcal{A}$ on a graph with edge weights $Cw_G$ will preserve a $C\varepsilon$-DP guarantee with respect to all graphs which are formed by starting with $Cw_G$ and iteratively changing $C$ edge weights by 1. In particular, this holds for $Cw_{G'}$. Rescaling the solution by $1/C$ yields a reduction of error by a factor of $1/C$ at the cost of increasing the privacy parameter by a factor of $C$. As the error of $\mathcal{A}$ scales linearly in $1/\varepsilon$, rescaling the privacy parameter by a factor of $C$ yields an equivalent error/privacy tradeoff for the $\ell_1$ notion of neighboring graphs.

We note that a yet more restrictive notion of privacy is used for privacy in the context of shortest path problems, e.g., [81, 22]. In the context of shortest paths, a zero-weight edge is not the same as a non-edge: a non-edge is equivalent to an edge of infinite weight. Therefore, standard notions of edge differential privacy do not allow for any approximation to path lengths. The notion considered in these shortest path problems is edge *weight* differential privacy, where the unweighted topology of the graph is fixed and made public while the weights on each edge are private. Specifically, neighboring graphs have the same edge set $E$ but differ in weights on those edges in $\ell_1$ distance at most 1. This is in contrast to the former notion of edge differential privacy, where two graphs are neighboring if one contains an edge of weight 1 and the other has a non-edge in that location.

## B    Private Min Isolating Cuts

In this section we prove Theorem 2.1. In fact, we prove a stronger version given by Lemma B.1. The steps in the algorithm that differ meaningfully from the non-private version are in color. We refer to Appendix A.1 and Table 2 for the definitions of our notation.

---

**Algorithm 1:** PrivateMinIsolatingCuts($G = (V, E, w), R, U, \varepsilon, \beta$)

**1** Initialize $W_r \leftarrow V$ for every $r \in R$
**2** Identify $R$ with $\{0, \ldots, |R| - 1\}$
**3** **for** $i$ *from* 0 *to* $\lfloor \lg(|R| - 1) \rfloor$ **do**
**4**     $A_i \leftarrow \{r \in R : r \bmod 2^{i+1} < 2^i\}$
**5**     $C_i \leftarrow$ PrivateMin-S-T-Cut($G, A_i, R \setminus A_i, \varepsilon/(\lg |R| + 3), \beta/(\lg |R| + 3)$)
**6**     $W_r \leftarrow W_r \cap C_i$ for every $r \in A_i$
**7**     $W_r \leftarrow W_r \cap (V \setminus C_i)$ for every $r \in R \setminus A_i$
**8** **end**
**9** **for** $r \in R$ **do**
**10**     Let $H_r$ be $G$ with all vertices in $V \setminus W_r$ contracted, and let $t_r$ be the contracted vertex
**11**     In $H_r$, add weight $B_\mathrm{H} \cdot \frac{(n + \lg(1/\beta)) \lg^2(|R|)\}}{\varepsilon |U|}$ between every vertex in $W_r \cap U$ and $t_r$ for some sufficiently large constant $B_\mathrm{H}$
**12** **end**
**13** $\mathcal{H} \leftarrow \bigcup_{r \in R} H_r$
**14** $\mathcal{C} \leftarrow$ PrivateMin-S-T-Cut($\mathcal{H}, R, \{t_r\}_{r \in R}, \varepsilon/(\lg |R| + 3), \beta/(\lg |R| + 3)$)
**15** **return** $\{\mathcal{C} \cap W_r\}_{r \in R}$

---

**Lemma B.1.** *On a graph $G$ with $n$ vertices, a set of terminals $R \subseteq V$, another set of vertices $U \subseteq V$, and a privacy parameter $\varepsilon$, there is an $(\varepsilon, 0)$-DP algorithm PrivateMinIsolatingCuts($G, R, U, \varepsilon, \beta$) that returns a set of Isolating Cuts over terminals $R$. The total cut values of the Isolating Cuts is within additive error $O((n + \lg(1/\beta)) \lg^2(|R|)/\varepsilon)$ from the Min Isolating Cuts with probability $1 - \beta$.*

*Furthermore, suppose the Min Isolating Cut for any terminal $r \in R$ contains at most $0.5|U|$ vertices from $U$. In that case, the Isolating Cut for $r$ returned by the algorithm will, with probability $1 - \beta$, contain at most $0.9|U|$ vertices from $U$.*

*Proof.* The algorithm is presented in Algorithm 1. On a high level, the algorithm follows the non-private Min Isolating Cuts algorithm by [62, 3], but replacing all calls to Min-$S$-$T$-Cut with private Min-$S$-$T$-Cut from Corollary A.2. One added step is Line 10, which is used to provide the guarantee that if the Min Isolating Cut for a terminal $r \in R$ contains a small number of vertices in $U$, then the isolating cut for terminal $r$ returned by the algorithm also does.

Next, we explain the algorithm in more detail. For every $r \in R$, we maintain a set $W_r$ that should contain the $r$-side of a cut seperating $r$ from $R \setminus \{r\}$ obtained in the algorithm. In each of the $\lfloor \lg(|R| - 1) \rfloor + 1$ iterations, we find a subset $A_i \subseteq R$, and find a cut that separates $A_i$ from $R \setminus A_i$. Let $C_i$ be the side of the cut containing $A_i$. Then for every $r \in A_i$, we update $W_r$ with $W_r \cap C_i$; for every $r \in R \setminus A_i$, we update $W_r$ with $W_r \cap (V \setminus C_i)$. The choice of $A_i$ is so that every pair $r_1, r_2 \in R$ are on different sides of the Min-$S$-$T$-Cut in at least one iteration; as a result, $W_r \cap R = \{r\}$ for every $r \in R$ after all iterations.

Next, for every $r \in R$, the algorithm aims to compute a cut separating $r$ from $R \setminus \{r\}$, where the side containing $r$ is inside $W_r$. This can be done by contracting all vertices outside of $W_r$ to a vertex $t_r$ and computing private Min-$r$-$t_r$-Cut. To incentivize cuts that contain fewer vertices in $U$ on the side containing $r$, the algorithm adds an edge with positive weight from every vertex in $W_r \cap U$ to $t_r$. Finally, these private Min-$r$-$t_r$-Cut instances can be solved at once by combining them into a single graph $\mathcal{H}$.

**Privacy analysis.** Considering the first for loop in the algorithm, The only parts that depend on the edges or edge weights are the calls to PrivateMin-S-T-Cut. Each call to PrivateMin-S-T-Cut is $(\varepsilon/(\lg |R| + 3), 0)$-DP, and the number of calls is $\lfloor \lg(|R| - 1) \rfloor + 1$, so this part of the algorithm is $\frac{\log |R| + 1}{\log |R| + 3}\varepsilon$-DP via basic composition (Theorem A.2).

Next, note that the sets $W_r$ are private since they are obtained from postprocessing the $\lg |R|$ privately computed min cuts $C_i$. Furthermore, the sets $W_r$ form a partition of $V$. This implies that an edge in the initial graph contributes its weight to at most two edges in $\mathcal{H}$. Namely, an edge internal to some $W_r$ appears only in $H_r$ and an edge between some $W_r$ and $W_{r'}$ is only contracted into an edge in $H_r$ and an edge in $H_{r'}$. Thus, the sensitivity of $\mathcal{H}$ is 2, so running PrivateMin-S-T-Cut on $\mathcal{H}$ is $2\varepsilon/(\log |R| + 3$-DP. Hence, the overall algorithm is $\varepsilon$-DP.

**Error analysis.** First, we analyze the error introduced by the **for** loop starting at Line 3. Let $\{S_r\}_{r \in R}$ be the (non-private) Min Isolating Cuts for terminals in $R$, which are only used for analysis purposes. Recall by Fact A.1, we can assume $\{S_r\}_{r \in R}$ are disjoint. Take an iteration $i$ of the **for** loop and let $\{W_r\}_{r \in R}$ be the values of $W_r$'s before the start of the iteration, and let $\{W'_r\}_{r \in R}$ denote the value of $W_r$'s at the end of the iteration.

Recall that with probability $1 - \beta/(\lg(|R|) + 3)$, the PrivateMin-S-T-Cut algorithm Corollary A.2 has additive error $O((n + \lg((\lg(|R|) + 3)/\beta))(\lg(|R|) + 3)/\varepsilon) = O((n + \lg(1/\beta))\lg(|R|)/\varepsilon)$. Hence, in the following analysis, we assume all calls of the PrivateMin-S-T-Cut algorithm have additive error $O((n + \lg(1/\beta))\lg(|R|)/\varepsilon)$ (which holds with probability $1 - \beta$ by union bound).

We show the following claim:

**Claim 1.** *It holds that*
$$\sum_{r \in R} w\left(W'_r \cap S_r\right) \leq \sum_{r \in R} w\left(W_r \cap S_r\right) + O((n + \lg(1/\beta))\lg(|R|)/\varepsilon).$$

*Proof.* We first show $\sum_{r \in A_i} w\left(W'_r \cap S_r\right) \leq \sum_{r \in A_i} w\left(W_r \cap S_r\right) + O(n \lg(|R|)/\varepsilon)$. Let $\mathbb{S}_{A_i} := \bigcup_{r \in A_i}(W_r \cap S_r)$. By Lemma A.1,
$$w(\mathbb{S}_{A_i}) + w(C_i) \geq w(\mathbb{S}_{A_i} \cup C_i) + w(\mathbb{S}_{A_i} \cap C_i). \tag{1}$$
Recall that with probability $1 - \beta/(\lg(|R|) + 3)$, $C_i$ is within $O((n + \lg((\lg(|R|) + 3)/\beta))(\lg(|R|) + 3)/\varepsilon) = O((n + \lg(1/\beta))\lg(|R|)/\varepsilon)$ of the minimum cut separating $A_i$ and $R \setminus A_i$, by the guarantee of Corollary A.2, and note that $\mathbb{S}_{A_i} \cup C_i$ is also a cut separating $A_i$ and $R \setminus A_i$. Therefore,
$$w(C_i) \leq w(\mathbb{S}_{A_i} \cup C_i) + O((n + \lg(1/\beta))\lg(|R|)/\varepsilon). \tag{2}$$
Combining Equations (1) and (2), we get that
$$w(\mathbb{S}_{A_i} \cap C_i) \leq w(\mathbb{S}_{A_i}) + O((n + \lg(1/\beta))\lg(|R|)/\varepsilon). \tag{3}$$

Therefore,

$$\sum_{r\in A_i} w\left(W'_r \cap S_r\right) = \sum_{r\in A_i} w\left(W_r \cap S_r \cap C_i\right)$$

$$= w\left(\bigcup_{r\in A_i}\left(W_r \cap S_r \cap C_i\right)\right) + \sum_{r_1\neq r_2\in A_i} w\left(E \cap \left(\left(W_{r_1}\cap S_{r_1}\cap C_i\right)\times\left(W_{r_2}\cap S_{r_2}\cap C_i\right)\right)\right)$$

$$\leq w(\mathbb{S}_{A_i}\cap C_i) + \sum_{r_1\neq r_2\in A_i} w\left(E \cap \left(\left(W_{r_1}\cap S_{r_1}\right)\times\left(W_{r_2}\cap S_{r_2}\right)\right)\right)$$

$$\leq w(\mathbb{S}_{A_i}) + O((n+\lg(1/\beta))\lg(|R|)/\varepsilon) + \sum_{r_1\neq r_2\in A_i} w\left(E \cap \left(\left(W_{r_1}\cap S_{r_1}\right)\times\left(W_{r_2}\cap S_{r_2}\right)\right)\right)$$

$$\text{(by Equation (3))}$$

$$= \sum_{r\in A_i} w\left(W_r \cap S_r\right) + O((n+\lg(1/\beta))\lg(|R|)/\varepsilon).$$

By an analogous argument, we can show $\sum_{r\in R\setminus A_i} w\left(W'_r \cap S_r\right) \leq \sum_{r\in R\setminus A_i} w\left(W_r \cap S_r\right) + O(n\lg(|R|)/\varepsilon)$. Summing up the two inequalities gives the desired claim. $\qquad\square$

By applying Claim 1 repeatedly, we can easily show the following claim:

**Claim 2.** *At the end of the **for** loop starting at Line 3,* $\sum_{r\in R} w(W_r\cap S_r) \leq \sum_{r\in R} w(S_r) + O((n+\lg(1/\beta))\lg^2(|R|)/\varepsilon)$.

*Proof.* Before the **for** loop starting at Line 3, we have $W_r = V$ for every $r \in R$, so $\sum_{r\in R} w(W_r \cap S_r) = \sum_{r\in R} w(S_r)$. Claim 1 shows that after each iteration of the **for** loop, the quantity $\sum_{r\in R} w(W_r \cap S_r)$ does not increase by more than $O((n+\lg(1/\beta))\lg(|R|)/\varepsilon)$. As there are $O(\lg(|R|))$ iterations, we get that at the end of the **for** loop,

$$\sum_{r\in R} w(W_r \cap S_r) \leq \sum_{r\in R} w(S_r) + O((n+\lg(1/\beta))\lg^2(|R|)/\varepsilon).$$

$\qquad\square$

The following claim is a simple observation:

**Claim 3.** *At the end of the **for** loop starting at Line 3, $r \in W_r$ for every $r \in R$ and distinct $W_r$'s are disjoint.*

Next, we show that the Min-$r$-$t_r$-Cut values in $H_r$ are close to the Min Isolating Cut values:

**Claim 4.** $\sum_{r\in R}\lambda_{H_r}(r,t_r) \leq \sum_{r\in R} w_G(S_r) + O((n+\lg(1/\beta))\lg^2(|R|)/\varepsilon)$.

*Proof.* We have that

$$\sum_{r\in R}\lambda_{H_r}(r,t_r) \leq \sum_{r\in R} w_{H_r}(W_r\cap S_r)$$

$$= \sum_{r\in R}\left(w_G(W_r\cap S_r) + |W_r\cap S_r\cap U|\cdot O((n+\lg(1/\beta))\lg^2(|R|)/(\varepsilon|U|))\right)$$

$$\leq \left(\sum_{r\in R} w_G(W_r\cap S_r)\right) + O((n+\lg(1/\beta))\lg^2(|R|)/\varepsilon)$$

$$\text{(as } \{S_r\}_{r\in R} \text{ are disjoint)}$$

$$\leq \left(\sum_{r\in R} w_G(S_r)\right) + O((n+\lg(1/\beta))\lg^2(|R|)/\varepsilon). \qquad\text{(by Claim 2)}$$

$\qquad\square$

Because of Claim 4, and the guarantee of Corollary A.2, the final cuts $\mathcal{C} \cap W_r$ returned by the algorithm will have the property that

$$\sum_{r \in R} w_{H_r}(\mathcal{C} \cap W_r) \leq \sum_{r \in R} \lambda_{H_r}(r, t_r) + O((n + \lg(1/\beta)) \lg(|R|)/\varepsilon)$$

$$\leq \sum_{r \in R} w_G(S_r) + O((n + \lg(1/\beta)) \lg^2(|R|)/\varepsilon).$$

Furthermore, $w_G(\mathcal{C} \cap W_r) \leq w_{H_r}(\mathcal{C} \cap W_r)$ as we only add positive weights to $H_r$ compared to $G$, we further get

$$\sum_{r \in R} w_G(\mathcal{C} \cap W_r) \leq \sum_{r \in R} w_G(S_r) + O((n + \lg(1/\beta)) \lg^2(|R|)/\varepsilon),$$

which is the desired error bound.

**Additional guarantee.** Finally, we need to show that if $|S_r \cap U| \leq 0.5U$ for some $r \in R$, then with probability $1 - \beta$, the returned isolating cut by the algorithm $\mathcal{C} \cap W_r$ has $|\mathcal{C} \cap W_r \cap U| \leq 0.9U$. Again, we can assume all calls of the PrivateMin-S-T-Cut algorithm have additive error $O((n + \lg(1/\beta)) \lg(|R|)/\varepsilon)$. As $\mathcal{C}$ is a cut returned by the PrivateMin-S-T-Cut algorithm separating $R$ and $\{t_r\}_{r \in R}$ and $(\mathcal{C} \setminus W_r) \cup (S_r \cap W_r)$ is also a cut separating $R$ and $\{t_r\}_{r \in R}$, we have

$$w_{\mathcal{H}}(\mathcal{C}) \leq w_{\mathcal{H}}((\mathcal{C} \setminus W_r) \cup (S_r \cap W_r)) + O((n + \lg(1/\beta))(\lg(|R|))/\varepsilon).$$

By removing cut values contributed by $H_{r'}$ for $r' \neq r$ from both sides, we get that

$$w_{H_r}(\mathcal{C} \cap W_r) \leq w_{H_r}(S_r \cap W_r) + O((n + \lg(1/\beta))(\lg(|R|))/\varepsilon).$$

Rewriting the cut values in terms of the edge weights of $G$ instead of $H_r$ (recall the weights of $G$ and $H_r$ are related as shown in Line 11), the above becomes

$w(\mathcal{C} \cap W_r) + (B_{\mathrm{H}} \cdot (n + \lg(1/\beta)) \lg^2(|R|)/(\varepsilon|U|))|\mathcal{C} \cap W_r \cap U|$
$\leq w(S_r \cap W_r) + (B_{\mathrm{H}} \cdot (n + \lg(1/\beta)) \lg^2(|R|)/(\varepsilon|U|))|S_r \cap W_r \cap U| + O((n + \lg(1/\beta))(\lg(|R|))/\varepsilon)$
$\leq w(S_r) + (B_{\mathrm{H}} \cdot (n + \lg(1/\beta)) \lg^2(|R|)/(\varepsilon|U|))|S_r \cap W_r \cap U| + O((n + \lg(1/\beta))(\lg^2(|R|))/\varepsilon).$
(By Claim 2)

Because $w(\mathcal{C} \cap W_r) \geq w(S_r)$ by definition of $S_r$, the above implies

$$|\mathcal{C} \cap W_r \cap U| \leq \frac{O((n + \lg(1/\beta))(\lg^2(|R|))/\varepsilon)}{B_{\mathrm{H}} \cdot (n + \lg(1/\beta)) \lg^2(|R|)/(\varepsilon|U|)} + |S_r \cap W_r \cap U|$$

$$\leq 0.4|U| + |S_r \cap W_r \cap U| \qquad \text{(By setting } B_{\mathrm{H}} \text{ large enough)}$$

$$\leq 0.4|U| + |S_r \cap U|$$

$$\leq 0.9|U|,$$

as desired. $\qquad\square$

## C  Core Recursive Step

We now describe a key subroutine, outlined as Algorithm 2, used to compute a DP Gomory-Hu tree. The high-level goal is to use Min Isolating Cuts to find minimum cuts that cover a large fraction of vertices in the graph. The overall structure of this algorithm follows that of the prior work [61] with several key changes to handle additive approximations and privacy. The inputs to Algorithm 2 are the weighted graph, a source vertex $s$, a set of active vertices $U \subseteq V$, a privacy parameter $\varepsilon$, and a failure probability $\beta$. The steps that differ meaningfully from the non-private version developed in [61] are in color. To obtain a DP version of this method, Algorithm 2 invokes the DP Min-$s$-$t$-Cut and the DP Min Isolating Cuts algorithm; the latter primitive is developed in this work in Appendix B. In the original non-private algorithm, isolating cuts $S_v^i$ are included in $D^i$ if the set $S_v^i$ corresponds to the $v$ side of the Min-$s$-$v$-Cut, i.e., $w(S_v^i) = \lambda(s, v)$. The analysis in prior work relies on this equality, i.e., on $w(S_v^i)$ and $\lambda(s, v)$ being the same, in a crucial way. Informally speaking, it enables the selection of many Min Isolating Cuts of the right size. In our case, since the cuts and their values are released

---

**Algorithm 2:** PrivateGHTreeStep($G = (V, E, w), s, U, \varepsilon, \beta$)

---

**1** $\Gamma_{\text{iso}} \leftarrow O\left(\frac{(n + \lg(1/\beta)) \lg^2(|U|)}{\varepsilon}\right)$ and $\Gamma_{\text{values}} \leftarrow O\left(\frac{|U| \lg(|U|/\beta)}{\varepsilon}\right)$

**2** $\hat{\lambda}(s, v) \leftarrow \lambda(s, v) + \text{Lap}\left(\frac{4(|U|-1)}{\varepsilon}\right)$ for all $v \in U \setminus \{s\}$

**3** Initialize $R^0 \leftarrow U$

**4 for** *i from* 0 *to* $\lfloor \lg |U| \rfloor$ **do**

**5**      Call PrivateMinIsolatingCuts$\left(G, R^i, U, \frac{\varepsilon}{2(\lfloor \lg |U| \rfloor + 1)}, \frac{\beta}{\lfloor \lg |U| \rfloor + 1}\right)$ (Algorithm 1) obtaining

      disjoint sets $\hat{S}_v^i$ ;                            /* $v$ ranges over vertices in $R^i$ */

**6**      $\hat{w}(\hat{S}_v^i) \leftarrow w(\hat{S}_v^i) + \text{Lap}\left(\frac{8(\lfloor \lg |U| \rfloor + 1)}{\varepsilon}\right)$ for each $v \in R^i \setminus \{s\}$

**7**      Let $D^i \subseteq U$ be the union of $\hat{S}_v^i \cap U$ over all $v \in R^i \setminus \{s\}$ satisfying

      $\hat{w}(\hat{S}_v^i) \leq \hat{\lambda}(s, v) + (2(\lfloor \lg |U| \rfloor - i) + 1)\Gamma_{\text{iso}} + \Gamma_{\text{values}}$ and $|\hat{S}_v^i \cap U| \leq (9/10)|U|$

**8**      $R^{i+1} \leftarrow$ sample of $U$ where each vertex in $U \setminus \{s\}$ is sampled independently with

      probability $2^{-i+1}$, and $s$ is sampled with probability 1

**9 end**

**10 return** $D$ (the largest set $D^i$), $R$ (the set of terminals $v \in R^i \setminus \{s\}$ satisfying the conditions on

     Line 7), and sets $\hat{S}_v^i$ for $v \in R$.

---

privately by random perturbations, it is unclear how to test that condition with equality. On the other hand, we still would like to ensure that many isolating cuts have "the right" size. Among our key technical contributions is relaxing that condition by using a condition which **changes** from iteration to iteration of the for-loop. The actual condition we use is

$$\hat{w}(\hat{S}_v^i) \leq \hat{\lambda}(s, v) + (2(\lfloor \lg |U| \rfloor - i) + 1)\Gamma_{\text{iso}} + \Gamma_{\text{values}} \tag{4}$$

on Line 7 of Algorithm 2 where $\Gamma_{\text{iso}}$ and $\Gamma_{\text{values}}$ are upper bounds on the additive errors of the approximate Min Isolating Cuts and the approximate Min-$s$-$v$-Cut values, respectively.

When using Equation (4), we also have to ensure that significant progress can still be made, i.e., to ensure that both (a) we will find a large set $D^i$ which is the union of approximate Min Isolating Cuts $\hat{S}_v^i$ satisfying the condition above and (b) none of the individual $\hat{S}$ which we return are too large as we will recurse within each of these sets. A new analysis uses this changing inequality to show that the former is true. For the latter, we utilize the special property of our PrivateMinIsolatingCuts in Appendix B that forces an approximate isolating cut to contain at most $0.9|U|$ terminals if there exists an exact isolating cut of size at most $|U|/2$. We now turn to the analysis.

### C.1 Correctness

As in prior work [61, 4], let $D^* \subseteq U \setminus \{s\}$ be the set of vertices $v$ such that if $S_v^*$ is the $v$ side of the Min-$s$-$v$-Cut, $|S_v^* \cap U| \leq |U|/2$.

**Lemma C.1.** *PrivateGHTreeStep($G, U, s, \varepsilon, \beta$) (Algorithm 2) has the following properties:*

- *Let $\Gamma_{iso} = C_1(n + \lg(1/\beta)) \lg^2(|U|)/\varepsilon$ and $\Gamma_{values} = C_2|U| \lg(|U|/\beta)/\varepsilon$ for large enough constants $C_1, C_2$. Let $\{S_v^i\}_{v \in R^i}$ be the optimal Min Isolating Cuts for terminals $R^i$ (by Fact A.1, these are disjoint without loss of generality). Let $R^*$ be the set of vertices $v$ for which Algorithm 2 returns $\hat{S}_v^i$. Then with probability at least $1 - O(\beta)$, the sets $\{\hat{S}_v^i : v \in R^*\}$ returned by Algorithm 2 are approximate Min Isolating Cuts and approximate Min-$v$-$s$-Cuts:*

$$\sum_{v \in R^*} w(\hat{S}_v^i) - w(S_v^i) \leq \Gamma_{iso},$$

*and, for all $v \in R^*$,*

$$w(\hat{S}_v^i) - \lambda(s, v) \leq 2(\lfloor \lg |U| \rfloor + 1)\Gamma_{iso} + 2\Gamma_{values}.$$

- *$D$ returned by the algorithm satisfies*

$$\mathbb{E}[|D|] = \Omega\left(\frac{|D^*|}{\lg |U|}\right).$$

To prove this, we will need the following helpful definition and lemma. Let $X_v^i$ be a random variable for the number of vertices in $U$ added to $D^i$ by a set $\hat{S}_v^i$:

$$X_v^i = \begin{cases} |\hat{S}_v^i \cap U| & \text{if } v \in R^i \text{ and } |\hat{S}_v^i \cap U| \leq (9/10)|U| \text{ and} \\ & \quad \hat{w}(\hat{S}_v^i) \leq \hat{\lambda}(s,v) + (2(\lfloor \lg |U| \rfloor - i) + 1)\Gamma_{\text{iso}} + \Gamma_{\text{values}} \, . \\ 0 & \text{o.w.} \end{cases} \tag{5}$$

**Lemma C.2.** *Consider a vertex $v \in D^*$. Let $S_v^i$ be the $v$ part of the optimal solution to Min Isolating Cuts at stage $i$. Assume that $|\lambda(s,v) - \hat{\lambda}(s,v)| \leq \Gamma_{values}$ and $|w(\hat{S}_v^i) - \hat{w}(\hat{S}_v^i)| + |w(S_v^i) - w(\hat{S}_v^i)| \leq \Gamma_{iso}$ for all $i \in \{0, \dots \lfloor \lg |U| \rfloor\}$. Additionally, if $|S_v^i \cap U| \leq |U|/2$, assume $\hat{S}_v^i \leq (9/10)|U|$. Then, there exists an $i' \in \{0, \dots \lfloor \lg |U| \rfloor\}$ such that*

$$\mathbb{E}\left[ X_v^{i'} \right] = \Omega(1).$$

*Proof.* Consider a specific sampling level $i \in \{0, \dots, \lfloor \lg |U| \rfloor\}$. We say that $i$ is "active" if there exists a set $\tilde{S}_v^i \subset U$ containing $v$ and not $s$ such that $|\tilde{S}_v^i \cap U| \in [2^i, 2^{i+1})$ and

$$w(\tilde{S}_v^i) \leq \hat{\lambda}(s,v) + 2(\lfloor \lg(|U|) \rfloor - i)\Gamma_{\text{iso}} + \Gamma_{\text{values}}. \tag{6}$$

Note that this is a deterministic property regarding the existence of such a set $\tilde{S}_v^i$ independent of the randomness used to sample terminals or find private Min Isolating Cuts.

Let $i'$ be the smallest active $i$. Recall $S_v^*$ is the $v$ side of the true Min-$s$-$v$-Cut. Let $i^* = \lfloor \lg |S_v^* \cap U| \rfloor$. As $w(S_v^*) = \lambda(s,v) \leq \hat{\lambda}(s,v) + \Gamma_{\text{values}}$, $i^*$ must be active, so $i'$ is well-defined and $i' \leq i^*$. As $i'$ is active, there exists a set $\tilde{S}_v^{i'}$ with $|\tilde{S}_v^{i'} \cap U| \in [2^{i'}, 2^{i'+1})$ and with cost within $2(\lfloor \lg(|U|) \rfloor - i')\Gamma_{\text{iso}} + \Gamma_{\text{values}}$ of $\hat{\lambda}(s,v)$. On the other hand, as $i'$ is the *smallest* active level, there is no set containing $v$ but not $s$, whose intersection with $U$ is less than $2^{i'}$, and whose cost is within $2(\lfloor \lg(|U|) \rfloor - (i' - 1))\Gamma_{\text{iso}} + \Gamma_{\text{values}}$ of $\hat{\lambda}(s,v)$.

Consider the event that in $R^{i'}$, we sample $v$ but no other vertices in $\tilde{S}_v^{i'}$ as terminals, i.e., $R^{i'} \cap \tilde{S}_v^{i'} = \{v\}$. Then, $\tilde{S}_v^{i'}$ is a cut separating $v$ and $R^{i'} \setminus \{v\}$. By the assumed guarantee of in the lemma statement, the actual cut we output has approximated cost:

$$\begin{aligned} \hat{w}(\hat{S}_v^{i'}) &\leq w(S_v^{i'}) + |w(S_v^{i'}) - \hat{w}(\hat{S}_v^{i'})| \\ &\leq w(\tilde{S}_v^{i'}) + |w(S_v^{i'}) - \hat{w}(\hat{S}_v^{i'})| \\ &\leq w(\tilde{S}_v^{i'}) + |w(S_v^{i'}) - w(\hat{S}_v^{i'})| + |w(\hat{S}_v^{i'}) - \hat{w}(\hat{S}_v^{i'})| \\ &\leq w(\tilde{S}_v^{i'}) + \Gamma_{\text{iso}} \\ &\leq \hat{\lambda}(s,v) + (2(\lfloor \lg(|U|) \rfloor - i') + 1)\Gamma_{\text{iso}} + \Gamma_{\text{values}}. \end{aligned}$$

For sake of contradiction, assume that $|\hat{S}_v^{i'} \cap U| < 2^{i'}$. Using the fact that all solutions of this size have large cost, we can conclude that

$$\begin{aligned} \hat{w}(\hat{S}_v^{i'}) &\geq w(\hat{S}_v^{i'}) - \Gamma_{\text{iso}} \\ &> \hat{\lambda}(s,v) + 2(\lfloor \lg(|U|) \rfloor - (i' - 1))\Gamma_{\text{iso}} + \Gamma_{\text{values}} - \Gamma_{\text{iso}} \\ &> \hat{\lambda}(s,v) + (2(\lfloor \lg(|U|) \rfloor - i')) + 1)\Gamma_{\text{iso}} + \Gamma_{\text{values}}. \end{aligned}$$

This contradicts the previous inequality that shows that $\hat{w}(\hat{S}_v^{i'})$ is upper bounded by this quantity, so $|\hat{S}_v^{i'} \cap U| \geq 2^{i'}$ as long as the sampling event occurs.

Next, we show that $|\hat{S}_v^{i'} \cap U| \leq (9/10)|U|$. As $v \in D^*$, the true minimum cut $S_v^*$ has the property $|S_v^* \cap U| \leq |U|/2$. Furthermore, by the isolating cuts lemma of [62], there is one minimum isolating cut solution for any set of terminals including $v$ and $s$ will have that the $v$ part $S_v$ is a subset of $S_v^*$ (this is the basis for the isolating cuts algorithm). So, if $v$ is sampled in $R^i$, there will exist an optimal isolating cuts solution $S_v^i$ with $|S_v^i \cap U| \leq |U|/2$. By the assumption in the lemma statement, $|\hat{S}_v^{i'} \cap U| \leq (9/10)|U|$.

Overall, we can bound the contribution of $\hat{S}_v^{i'}$ to $D^{i'}$ as

$$\mathbb{E}\left[X_v^{i'}\right] \geq |S_v^{i'} \cap U| \cdot \mathbb{P}\left[v \text{ is the only vertex sampled in } \tilde{S}_v^{i'} \text{ under sampling probability } 2^{-i'}\right]$$

$$\geq 2^{i'}\left(2^{-i'}\right)\left(1-2^{-i'}\right)^{|\tilde{S}_v^{i'}|-1}$$

$$\geq \left(1-2^{-i'}\right)^{2^{i'+1}-2}.$$

If $i' = 0$, this evaluates to 1. Otherwise, if $i' \geq 1$,

$$\mathbb{E}\left[X_v^{i'}\right] \geq \left(1-2^{-i'}\right)^{2^{i'+1}} = \left(\frac{1}{1+\frac{2^{-i'}}{1-2^{-i'}}}\right)^{2^{i'+1}} \geq \left(\frac{1}{e^{\frac{2^{-i'}}{1-2^{-i'}}}}\right)^{2^{i'+1}} = e^{-\frac{2}{1-2^{-i'}}} \geq e^{-4}.$$

$\square$

We are now ready to prove the main lemma of this section.

**Proof of Lemma C.1.** The first step of the proof will be to show that $\Gamma_{\text{iso}}$ and $\Gamma_{\text{values}}$ upper-bound the error of the approximate isolating cuts and min cut values used in the algorithm with probability $1 - O(\beta)$. Applying the guarantee of Lemma B.1 and union bounding over all $i$, the following guarantee of the quality of $\hat{S}_v^i$ holds with probability $1 - \beta$. If $\{S_v^i\}$ are optimal Min Isolating Cuts for terminals $R^i$:

$$\sum_{v \in R^i} w(\hat{S}_v^i) - w(S_v^i) \leq O\left(\frac{(n + \lg(1/\beta))\lg^2(|R^i|)}{\varepsilon}\right).$$

We remark that $w(\hat{S}_v^i) \geq w(S_v^i)$ due to the optimality of $S_v^i$.

Now noting (from Line 6 of Algorithm 2) that $\hat{w}(\hat{S}_v^i) - w(\hat{S}_v^i)$ is a Laplace random variable $\text{Lap}\left(\frac{8(\lfloor \lg |U| \rfloor + 1)}{\varepsilon}\right)$ and that the absolute value of it is distributed as $\text{Exp}\left(\frac{\varepsilon}{8(\lfloor \lg |U| \rfloor + 1)}\right)$ [60].

Note that the expected value of $\sum_{v \in R^i} |w(\hat{S}_v^i) - \hat{w}(\hat{S}_v^i)|$ is $\frac{8|R^i|(\lfloor \lg |U| \rfloor + 1)}{\varepsilon} = O(\frac{n \log |U|}{\varepsilon})$. We then apply Corollary A.1 with $a = \frac{\varepsilon}{8(\lfloor \lg |U| \rfloor + 1)}$ and $T = \Theta(\frac{(n + \log(1/\beta)) \log |U|}{\varepsilon})$ so that

$$\mathbb{P}\left[\sum_{v \in R^i} |w(\hat{S}_v^i) - \hat{w}(\hat{S}_v^i)| \geq T\right] \leq \exp\left(-\Omega(aT)\right) = \exp\left(-\Omega(n + \log(1/\beta))\right) \leq 1 - \beta,$$

when the constant factor hidden in the bound for $T$ is sufficiently large.

Therefore, with probability $1 - O(\beta)$, for all $i \in \{0, \ldots, \lfloor \lg |U| \rfloor\}$, it holds

$$\sum_{v \in R^i} |w(S_v^i) - w(\hat{S}_v^i)| + |w(\hat{S}_v^i) - \hat{w}(\hat{S}_v^i)| \leq \Gamma_{\text{iso}}.$$

This satisfies the approximate Min Isolating Cuts guarantee of the lemma.

Similarly, the additional guarantee in Lemma C.2 (if $|S_v^i \cap U| \leq |U|/2$, assume $\hat{S}_v^i \leq (9/10)|U|$) also holds with probability $1 - O(\beta)$ by union bound.

For the approximate min cut values, by the tail of the Laplace distribution and a union bound, each $\hat{\lambda}(s, v)$ satisfies

$$|\lambda(s, v) - \hat{\lambda}(s, v)| \leq \Gamma_{\text{values}} = O\left(\frac{|U|\lg(|U|/\beta)}{\varepsilon}\right)$$

with probability $1 - \beta$. We condition on these events moving forward.

Next, we show that sets $\hat{S}_v^i$ are only included in our output if they are close to the min cut value $\lambda(s, v)$. Specifically, for any set returned by our algorithm:

$$\hat{w}(\hat{S}_v^i) \leq \hat{\lambda}(s, v) + (2(\lfloor \lg |U| \rfloor - i) + 1)\Gamma_{\text{iso}} + \Gamma_{\text{values}}.$$

Applying the error guarantees for $\hat{w}(\hat{S}_v^i)$, $\hat{S}_v^i$, and $\hat{\lambda}(s, v)$ derived above,

$$
\begin{aligned}
w(\hat{S}_v^i) &\leq \Gamma_{\text{iso}} + \hat{w}(\hat{S}_v^i) \\
&\leq \Gamma_{\text{iso}} + \hat{\lambda}(s, v) + (2(\lfloor \lg |U| \rfloor - i) + 1)\Gamma_{\text{iso}} + \Gamma_{\text{values}} \\
&\leq \Gamma_{\text{iso}} + (\lambda(s, v) + \Gamma_{\text{values}}) + (2(\lfloor \lg |U| \rfloor - i) + 1)\Gamma_{\text{iso}} + \Gamma_{\text{values}} \\
&\leq \lambda(s, v) + 2(\lfloor \lg |U| \rfloor + 1)\Gamma_{\text{iso}} + 2\Gamma_{\text{values}}.
\end{aligned}
$$

This completes the first part of the proof concerning the error of the returned sets. In the remainder, we focus on the cardinality of the output.

By definition of $X_v^i$, the size of $D^i$ is given by a sum over $X_v^i$:

$$
|D^i| = \sum_{v \in U \setminus \{s\}} X_v^i
$$

By linearity of expectation and as $D^* \subseteq U \setminus \{s\}$,

$$
\mathbb{E} \left[ \sum_{i=0}^{\lfloor \lg |U| \rfloor} |D^i| \right] \geq \sum_{i=0}^{\lfloor \lg |U| \rfloor} \sum_{v \in D^*} \mathbb{E}\left[ X_v^i \right].
$$

As we output the largest $D^i$ across all $i$, the output of our algorithm will have an expected size of at least

$$
\frac{1}{\lfloor \lg |U| \rfloor + 1} \sum_{i=0}^{\lfloor \lg |U| \rfloor} \sum_{v \in D^*} \mathbb{E}\left[ X_v^i \right].
$$

Via Lemma C.2 (note that the error condition holds with probability $1 - O(\beta)$ from the first part of this proof), the expected output size will be at least

$$
\Omega\left( \frac{|D^*|}{\lg |U|} \right).
$$

$\square$

## C.2   Privacy

We now analyze the privacy guarantee of our algorithm. A key technical observation behind our analysis is that the (approximate) isolating cuts found by Algorithm 2 are *disjoint* subsets of vertices, so any edge can only appear in at most 2 sets at any sampling level in the **for** loop starting at Line 4. This is formalized in the lemma below.

**Lemma C.3.** *PrivateGHTreeStep (Algorithm 2) is $\varepsilon$-DP.*

*Proof.* The algorithm PrivateGHTreeStep interacts with the sensitive edges only through calculations of approximate min cut values $\hat{\lambda}(s, v)$, approximate min isolating cut values $\hat{w}(\hat{S}_v^i)$, and calls to PrivateMinIsolatingCuts (Algorithm 1). Otherwise, the computation only deals with the vertices of the graph, which are public. The calculation of each of the $|U| - 1$ cut values is $\frac{\varepsilon}{4(|U|-1)}$-DP via the Laplace mechanism (Theorem A.3) as a change in any edge weight by 1 can affect a cut value by at most 1. By basic composition (Theorem A.2), the total privacy of these calls is

$$
(|U| - 1)\frac{\varepsilon}{4(|U| - 1)} = \frac{\varepsilon}{4}.
$$

Via the privacy of PrivateMinIsolatingCuts, each call to that subroutine is $\frac{\varepsilon}{2(\lfloor \lg |U| \rfloor + 1)}$-DP. By basic composition, the total privacy of these calls is

$$
(\lfloor \lg |U| \rfloor + 1)\left( \frac{\varepsilon}{2(\lfloor \lg |U| \rfloor + 1)} \right) = \frac{\varepsilon}{2}.
$$

Consider the vector $x^i \in \mathbb{R}^{|R^i|-1}$ where each entry in $x^i$ corresponds to $w(\hat{S}_v^i)$ for some $v \in R^i \setminus \{s\}$. At any sampling level $i$, the approximate isolating cuts $\hat{S}_v^i$ are disjoint. Therefore, a change in any

edge weight by 1 can change at most two coordinates of $x^i$ each at most by 1 (namely, the coordinates corresponding to the sets $\hat{S}_v^i$ which contain the endpoints of the edge). So, $x^i$ has $\ell_1$-sensitivity 2 relative to the edge weights of the graph. By the Laplace mechanism, release of all noised entries of $x^i$, given by $\hat{w}(\hat{S}_v^i)$, for any fixed $i$ is $\frac{\varepsilon}{4(\lfloor \lg |U| \rfloor + 1)}$-DP. Summing over all sampling levels via basic composition, these calls have privacy

$$(\lfloor \lg |U| \rfloor + 1)\left(\frac{\varepsilon}{4(\lfloor \lg |U| \rfloor + 1)}\right) = \frac{\varepsilon}{4}.$$

In total, this algorithm is $\varepsilon$-DP as $\frac{\varepsilon}{4} + \frac{\varepsilon}{2} + \frac{\varepsilon}{4} = \varepsilon$. $\qquad\square$

## D  Final Algorithm

---
**Algorithm 3:** PrivateGHTree($G = (V, E, w), \varepsilon$)
---
1 $(T, f) \leftarrow$ PrivateGHSteinerTree($G, V, \varepsilon, 0, n$)

2 Add $\mathrm{Lap}\left(\frac{2(n-1)}{\varepsilon}\right)$ noise to each edge in $T$

3 **return** $T$

---

---
**Algorithm 4:** PrivateGHSteinerTree($G = (V, E, w), U, \varepsilon, t, n_{\max}$)
---
1 $t_{\max} \leftarrow \Theta(\lg^2 n_{\max})$

2 **if** $t \geq t_{\max}$ **then**

3 $\quad$ **return abort** ;                          /* the privacy budget is exhausted */

4 **end**

5 $s \leftarrow$ uniformly random vertex in $U$

6 Call PrivateGHTreeStep($G, s, U, \frac{\varepsilon}{4t_{\max}}, \frac{1}{n_{\max}^3}$) to obtain $D, R \subseteq U$ and disjoint sets $\hat{S}_v$ for
$\quad v \in R$ (recall $D = \bigcup_{v \in R} \hat{S}_v \cap U$)

7 **for** *each $v \in R$* **do**

8 $\quad$ Let $G_v$ be the graph with vertices $V \setminus \hat{S}_v$ contracted to a single vertex $x_v$

9 $\quad$ Add edges with weight $\mathrm{Lap}\left(\frac{8t_{\max}}{\varepsilon}\right)$ from $x_v$ to every other vertex in $G_v$, truncating resulting
$\quad\quad$ edge weights to be at least 0

10 $\quad U_v \leftarrow \hat{S}_v \cap U$

11 $\quad$ If $|U_v| > 1$, recursively set $(T_v, f_v) \leftarrow$ PrivateGHSteinerTree($G_v, U_v, \varepsilon, t+1, n_{\max}$);
$\quad\quad$ otherwise, $T_v$ is a single node and $f_v$ is the identity map

12 **end**

13 Let $G_{\mathrm{large}}$ be the graph $G$ with (disjoint) vertex sets $\hat{S}_v$ contracted to single vertices $y_v$ for all
$\quad v \in R$

14 $U_{\mathrm{large}} \leftarrow U \setminus D$

15 If $|U_{\mathrm{large}}| > 1$, recursively set
$\quad (T_{\mathrm{large}}, f_{\mathrm{large}}) \leftarrow$ PrivateGHSteinerTree($G_{\mathrm{large}}, U_{\mathrm{large}}, \varepsilon, t+1, n_{\max}$); otherwise, $T_{\mathrm{large}}$ is a
$\quad$ single node and $f_{\mathrm{large}}$ is the identity map

16 **return** Combine($(T_{\mathrm{large}}, f_{\mathrm{large}}), \{(T_v, f_v) : v \in R\}, \{\hat{S}_v : v \in R\}$)

---

---
**Algorithm 5:** Combine($(T_{\mathrm{large}}, f_{\mathrm{large}}), \{(T_v, f_v) : v \in R\}, \{\hat{S}_v : v \in R\}$)
---
1 Construct $T$ by starting with the disjoint union $T_{\mathrm{large}} \cup \bigcup_{v \in R} T_v$ and for each $v \in R$, adding an
$\quad$ edge between $f_v(x_v) \in U_v$ and $f_{\mathrm{large}}(y_v) \in U_{\mathrm{large}}$

2 Construct $f : V \to U = U_{\mathrm{large}} \cup \bigcup_{v \in R} U_v$ by $f(v') = f_{\mathrm{large}}(v')$ if $v' \in V \setminus \bigcup_{v \in R} \hat{S}_v$ and
$\quad f(v') = f_v(v')$ if $v' \in \hat{S}_v$ for some $v \in R$

---

In this section, we present the algorithm PrivateGHTree (Algorithm 3) for constructing an $\varepsilon$-DP approximate Gomory-Hu tree and analyze its approximation error and privacy guarantees. The steps

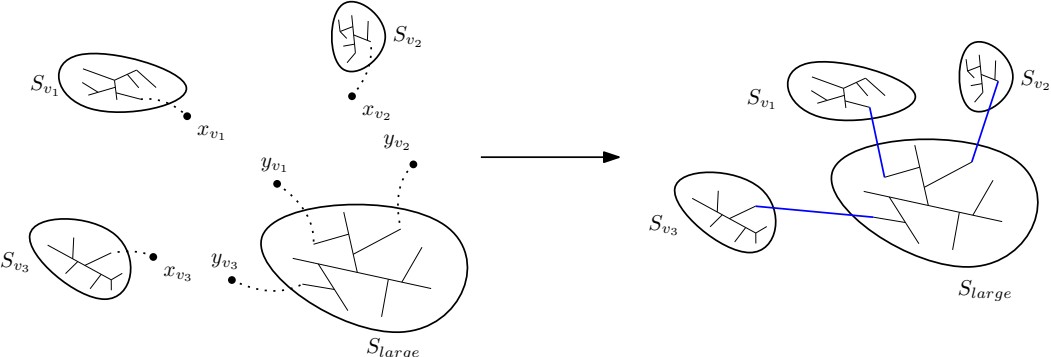

Figure 4: The Combine procedure of Algorithm 5. The computation on each recursive subinstance provides a Gomory-Hu Steiner tree on that subinstance. To stitch these together to a Gomory-Hu Steiner tree of the initial instance, we need to add edges between the solutions to the subinstances. For a node $v \in R$, such that $x_v$ is assigned to $f_v(x_v)$ in $S_v$ and $y_v$ is assigned to $f_{large}(y_v)$ in $S_{large}$ in the recursively obtained Gomory-Hu Steiner trees, we add an edge between $f_v(x_v)$ and $f_{large}(y_v)$.

that differ meaningfully from the non-private version developed in [61] are in color. As in [61], we construct the slightly more general structure of a *Gomory-Hu Steiner tree* as an intermediate step in Algorithm 4.

**Definition D.1.** *Let $G = (V, E, w)$ be a weighted graph and $U \subseteq V$ a set of terminals. A $\Gamma$-approximate Gomory-Hu Steiner tree is a weighted spanning tree $T$ on $U$ with a function $f : V \to U$ where $f|_U$ is the identity.*

*For all distinct $s, t \in U$, if $(u, v)$ is the minimum weight edge on the unique path between $s$ and $t$, in $T$, and if $U'$ is the connected component of $T \setminus \{(u, v)\}$ containing $s$, then $f^{-1}(U')$ is a $\Gamma$-approximate Min-s-t-Cut with $\lambda_G(s, t) \leq w_T(u, v) = w_G(f^{-1}(U')) \leq \lambda_G(s, t) + \Gamma$.*

To construct the final approximate Gomory-Hu tree, we make a call to PrivateGHSteinerTree (Algorithm 4) with $U = V$, the entire vertex set. The algorithm PrivateGHSteinerTree is a private version of the GHTree algorithm in [61]. It computes several (approximate) min cuts from a randomly sampled vertex $s \in U$ by making a call to PrivateGHTreeStep (Algorithm 2) to obtain $D, R \subseteq U$ and disjoint sets $\hat{S}_v$ (where $D = \bigcup_{v \in R} \hat{S}_v \cap U$). For each of these cuts $\hat{S}_v$ it constructs recursive sub-instances $(G_v, U_v)$ where $G_v$ is obtained by contracting $V \setminus \hat{S}_v$ to a single vertex $x_v$ and $U_v \leftarrow \hat{S}_v \cap U$. Moreover, it creates a sub-instance $(G_{large}, U_{large})$ by contracting each of $\hat{S}_v$ to a single vertex $y_v$ for $y \in R$ and setting $U_{large} = U \leftarrow D$.

Notably, on Line 8, where the algorithm recurses on the graph $G_v$ with $V \setminus \hat{S}_v$ contracted to a single vertex $x_v$, we add noisy edges from $x_v$ to all other vertices of the graph. This ensures the privacy of any actual edge from $x_v$ in the entire recursive subtree of that instance without incurring too much error. This will imply that for any edge and any instance during the recursion, there is at most one sub-instance where the edge does not receive this privacy guarantee. If $t$ is the depth of the recursion tree, this allows us to apply basic composition over only $O(t)$ computations of the algorithm. Essentially, there is only one path down the recursion tree on which we need to track privacy for any given edge in the original graph. We enforce $t < t_{max}$, and as we will show, the algorithm successfully terminates with depth less than $t_{max}$ with high probability.

To combine the solutions to the recursive sub-problems, we use the Combine algorithm (Algorithm 5) from [61], which in turn is similar to the original Gomory-Hu tree combine step except that it combines more than two recursive sub-instances. See Figure 4 for an illustration of the Combine step.

Finally, Algorithm 3 calls Algorithm 4 with privacy budget $\varepsilon/2$. To be able to output weights of the tree edges, it simply adds Laplace noise $\text{Lap}(\frac{2(n-1)}{\varepsilon})$ to value of the corresponding cuts in $G$, hence incurring error $O(\frac{n \lg n}{\varepsilon})$ with high probability. This also has privacy loss $\varepsilon/2$ by basic composition over the $n - 1$ tree edges, so the full algorithm is $\varepsilon$-differentially private.

## D.1 Correctness

In this section, we analyze the approximation guarantee of our algorithm. The following main lemma states that Algorithm 4 outputs an $O(n \operatorname{polylog}(n))$-approximate Gomory-Hu Steiner tree.

**Lemma D.1.** *Let* $t_{\max} = C \lg^2 n$ *for a sufficiently large constant* $C$. *PrivateGHSteinerTree$(G, V, \varepsilon, 0, n)$ outputs an* $O(\frac{n \lg^8 n}{\varepsilon})$-*approximate Gomory-Hu Steiner tree* $T$ *of* $G$ *with high probability.*

We start by proving a lemma for analyzing a single recursive step of the algorithm. It is similar to [61, Lemma 4.5.4] but its proof requires a more careful application of the submodularity lemma.

**Lemma D.2.** *With high probability, for any distinct vertices* $p, q \in U_{large}$, *we have that* $\lambda_G(p, q) \leq \lambda_{G_{large}}(p, q) \leq \lambda_G(p, q) + O(\frac{n \lg^5 n_{\max}}{\varepsilon})$. *Also with high probability, for any* $v \in R$ *and distinct vertices* $p, q \in U_v$, *we have that* $\lambda_G(p, q) \leq \lambda_{G_v}(p, q) \leq \lambda_G(p, q) + O(\frac{n \lg^6 n_{\max}}{\varepsilon})$.

*Proof.* Let us start by upper bounding how close the cuts $\hat{S}_v$ are to being Min-$s$-$v$-Cuts and how close the approximate min cut values $\hat{w}(\hat{S}_v)$ are to the true sizes $w(\hat{S}_v)$. Algorithm 4 calls Algorithm 2 with privacy parameter $\varepsilon_1 = \frac{\varepsilon}{4t_{\max}} = \Theta\left(\frac{\varepsilon}{\lg^2 n_{\max}}\right)$ and error parameter $\beta = \frac{1}{n_{\max}^3}$ where $n_{\max}$ is the number of vertices in the original graph. By Lemma C.1 with privacy $\varepsilon_1$ and error parameter $\beta$, it follows that (a) the sets $\{\hat{S}_v\}$ are approximate minimum isolating cuts with total error $O\left(\frac{(n+\log(1/\beta))\lg^2 n}{\varepsilon_1}\right) = O\left(\frac{n \lg^5 n_{\max}}{\varepsilon}\right)$ and (b) each set $\hat{S}_v$ is an approximate Min-$s$-$v$-Cut with error $O\left(\frac{n \lg^6 n_{\max}}{\varepsilon}\right)$. On any given call to Algorithm 2, these error bounds hold with probability $1 - n_{\max}^{-3}$. As each call to Algorithm 2 ultimately contributes an edge to the final Gomory-Hu tree via Algorithm 5, there can be at most $n_{\max} - 1$ calls throughout the entire recursion tree, resulting in failure probability of $n_{\max}^{-2}$ overall after a union bound.

The fact that $\lambda_G(p, q) \leq \lambda_{G_{large}}(p, q)$ follows since $G_{large}$ is a contraction of $G$. To prove the second inequality, let $S$ be one side of the true Min-$p$-$q$-Cut in $G$. Let $R_1 = R \cap S$ and $R_2 = R \setminus S$. We show that the cut $S^* = (S \cup \bigcup_{v \in R_1} \hat{S}_v) \setminus (\bigcup_{v \in R_2} \hat{S}_v)$ is an $O(n \lg^6 n_{\max}/\varepsilon)$-approximate min cut. Since $S^*$ is also a cut in $G_{large}$, the desired bound on $\lambda_{G_{large}}(p, q)$ follows.

Let $v_1, \ldots, v_{|R_1|}$ be the vertices of $R_1$ in an arbitrary order. By $|R_1|$ applications of the submodularity lemma (Lemma A.1),

$$w\left(S \cup \bigcup_{i=1}^{|R_1|} \hat{S}_{v_i}\right) \leq w(S) + \sum_{i=1}^{|R_1|} \left(w(\hat{S}_{v_i}) - w\left(\left(S \cup \bigcup_{j<i} \hat{S}_{v_j}\right) \cap \hat{S}_{v_i}\right)\right).$$

Note that $S \cup \bigcup_{i=1}^{|R_1|} \hat{S}_{v_i}$ is still a $(p, q)$-cut as $p, q \in U_{large}$ and the sets $\hat{S}_{v_i}$ are each disjoint from $U_{large}$. Moreover, for each $i$, $S$ contains $v_i$ and so $\left(S \cup \bigcup_{j<i} \hat{S}_{v_j}\right) \cap \hat{S}_{v_i}$ isolates $v_i$ from all vertices in $V \setminus \hat{S}_{v_i}$. Using the fact that $\{\hat{S}_v\}$ are approximate minimum isolating cuts, the sum in the RHS above can be upper bounded by $O(\frac{n \lg^5 n_{\max}}{\varepsilon})$. Letting $S' = S \cup \bigcup_{v \in R_1} \hat{S}_v$, and $S'' = (V \setminus S') \cup \bigcup_{v \in R_2} \hat{S}_v$ a similar argument but applied to $V \setminus S'$, shows that

$$w(S'') = w\left((V \setminus S') \cup \bigcup_{v \in R_2} \hat{S}_v\right) \leq w(V \setminus S') + O\left(\frac{n \lg^5 n_{\max}}{\varepsilon}\right).$$

But $S^* = V \setminus S''$, so we get that

$$w(S^*) = w(S'') \leq w(S') + O\left(\frac{n \lg^5 n_{\max}}{\varepsilon}\right) \leq w(S) + O\left(\frac{n \lg^5 n_{\max}}{\varepsilon}\right),$$

as desired.

For the case of $p, q \in U_v$ for some $v$, again the bound $\lambda_G(p, q) \leq \lambda_{G_u}(p, q)$ is clear. Thus, it suffices to consider the upper bound on $\lambda_{G_v}(p, q)$. Let $S$ be the side of the Min-$p$-$q$-Cut in $G$ which does not contain $v$. Assume first that $s \notin S$. By the submodularity lemma (Lemma A.1)

$$w(S \cap \hat{S}_v) \leq w(S) + w(\hat{S}_v) - w(S \cup \hat{S}_v).$$

By the approximation guarantees of Algorithm 2, $w(\hat{S}_v) \leq \lambda_G(s,v) + O(\frac{n \lg^6 n_{\max}}{\varepsilon})$. Moreover, note that $S \cup \hat{S}_v$ is an $(s,v)$-cut of $G$, so $w(S \cup \hat{S}_v) \geq \lambda_G(s,v)$. Thus,

$$w(S \cap \hat{S}_v) \leq w(S) + O\left(\frac{n \lg^6 n_{\max}}{\varepsilon}\right) = \lambda_G(p,q) + O\left(\frac{n \lg^6 n_{\max}}{\varepsilon}\right).$$

Since $S \cap \hat{S}_v$ is a $(p,q)$-cut of $G_v$, we must have that $w(S \cap \hat{S}_v) \geq \lambda_{G_v}(p,q)$, so in conclusion $\lambda_{G_v}(p,q) \leq \lambda_G(p,q) + O(\frac{n \lg^6 n_{\max}}{\varepsilon})$ *ignoring* the added noisy edges to $G_v$ in Line 8 of Algorithm 4.

Adding the noisy edges can only increase the cost by $O(\frac{n \lg^3 n_{\max}}{\varepsilon})$ with high probability via Laplace tail bounds (note that there can be at most $n_{\max} - 1$ noisy edges added as each time a noisy edge is added an edge is added to the final approximate Gomory Hu Steiner tree). This finishes the proof in the case $s \notin S$. A similar argument handles the case where $s \in S$ but here we relate the value $w(V \setminus S)$ to $w(V \setminus S) \cap \hat{S}_v$. $\qquad\square$

To bound the error of the algorithm, we need a further lemma bounding the depth of its recursion. The argument is similar to that of [61].

**Lemma D.3.** *If $t_{\max} = C \lg^2 n$ for a sufficiently large constant $C$, then, with high probability, no recursive call to Algorithm 4 from PrivateGHTree$(G, \varepsilon)$ aborts.*

*Proof.* Each of the recursive instances $(G_v, U_v)$ has $|U_v| \leq \frac{9}{10}|U|$ by the way $D$ is picked in Line 7 of Algorithm 2. Moreover, by [2, Corollary III.7], if $s$ is picked uniformly at random from $U$, then $\mathbb{E}[D^*] = \Omega(|U| - 1)$. By Lemma C.1, the expected size of $D$ returned by a call to Algorithm 2 when picking $s$ at random from $|U|$ is then at least $\Omega\left(\frac{|U|-1}{\lg n}\right)$. By Line 14 of Algorithm 4, it follows that, $\mathbb{E}[|U_{\text{large}}|] \leq |U|(1 - \Omega(1/\lg n))$ when $|U| > 1$. Thus any instance at recursive depth $t$ satisfies $\mathbb{E}[|U_{\text{large}}|] \leq n(1 - \Omega(1/\lg n))^t \leq n \exp(-\Omega(t/\lg n))$. If $t = \Omega(\lg^2 n)$, the expectation is smaller than $1/\operatorname{poly}(n)$, so by Markov's inequality, any instance at recursive depth $t$ satisfies $|U| = 1$ with high probability. Now note that at every recursive depth, we can only have at most $n$ instances, since the sets $U$ passed to the recursive calls of Algorithm 4 at depth $t$ are disjoint. Thus, there are polynomially many recursive instances up to recursive depth $t_{\max}$, so we can union bound over all sub-instances. In conclusion, all sub-instances have $|U| = 1$ within $O(\lg^2 n)$ recursive depth with high probability. $\qquad\square$

We can now prove Lemma D.1. The argument is again similar to [61] except we have to incorporate the approximation errors.

**Proof of Lemma D.1.** By Lemma D.3, the algorithm does not abort with high probability.

Throughout the proof, $n$ denotes the number of vertices in the input graph. Let $\Delta = O(\frac{n \lg^6 n}{\varepsilon})$ be such that with high probability $\lambda_{G_v}(p,q) \leq \lambda_G(p,q) + \Delta$ for $p,q \in U_v$ and similarly $\lambda_{G_{large}}(p,q) \leq \lambda_G(p,q) + \Delta$ for $p,q \in U_{large}$. The existence of $\Delta$ is guaranteed by Lemma D.2. We prove by induction on $i = 0, \ldots, t_{\max}$, that the output to the instances at level $t_{\max} - i$ of the recursion are $2i\Delta$-approximate Gomory-Hu Steiner trees. This holds trivially for $i = 0$ as the instances on that level have $|U| = 1$ and the tree is the trivial one-vertex tree approximating no cuts at all. Let $i \geq 1$ and assume inductively that the result holds for smaller $i$. In particular, if $(T, f)$ is the output of an instance at recursion level $i$, then the trees $(T_v, f_v)$ and $(T_{\text{large}}, f_{\text{large}})$ are $2(i-1)\Delta$-approximate Gomory-Hu Steiner trees of their respective $G_v$ or $G_{\text{large}}$ graphs.

Consider any internal edge $(a,b) \in T_{\text{large}}$ (without loss of generality, what follows also holds for $(a,b) \in T_v$). Let $U'$ and $U'_{\text{large}}$ be the connected component containing $a$ after removing $(a,b)$ from $T$ and $T_{\text{large}}$, respectively. By design of Algorithm 5, $f_{\text{large}}^{-1}(U'_{\text{large}})$ and $f^{-1}(U')$ are the same except each contracted vertex $y_v \in f_{\text{large}}^{-1}(U'_{\text{large}})$ appears as $\hat{S}_v \subseteq f^{-1}(U')$. It follows that $w_{G_{\text{large}}}(f_{\text{large}}^{-1}(U'_{\text{large}})) = w_G(f^{-1}(U'))$. By the inductive hypothesis, $(T_{\text{large}}, f_{\text{large}})$ is an approximate Gomory-Hu Steiner tree, so $w_{G_{\text{large}}}(f_{\text{large}}^{-1}(U'_{\text{large}})) = w_{T_{\text{large}}}(a,b)$. Therefore setting $w_T(a,b) = w_{T_{\text{large}}}(a,b) = w_G(f^{-1}(U'))$ has the correct cost for $T$ according to the definition of an approximate Gomory-Hu Steiner tree.

Furthermore, on the new edges $(f_v(x_v), f_{\text{large}}(y_v))$, the weight $w(\hat{S}_v)$ is the correct weight for that edge in $T$ as $\hat{S}_v$ is the $f_v(x_v)$ side of the connected component after removing that edge. Finally, by

adding these new edges, the resulting tree is a spanning tree. So, the structure of the tree is correct, and it remains to argue that the cuts induced by the tree (via minimum edges on shortest paths) are approximate Min-$s$-$t$-Cuts.

Consider any $p, q \in U$. Let $(a, b)$ be the minimum edge on the shortest path in $T$. Note that it is always the case that $w_T(a, b) \geq \lambda(a, b)$ as $w_T(a, b)$ corresponds to the value of a cut in $G$ separating $a$ and $b$. We will proceed by cases.

If $(a, b) \in T_{\text{large}}$, then by induction, $w_T(a, b) = w_{T_{\text{large}}}(a, b) \leq \lambda_{G_{\text{large}}}(a, b) + (2i - 2)\Delta$. By Lemma D.2, it follows that $w_T(a, b) \leq \lambda_G(a, b) + (2i - 1)\Delta$. The exact same argument applies if $(a, b) \in T_v$ for some $v \in R$.

The case that remains is if $(a, b)$ is a new edge with $a = f_v^{-1}(x_v) \in U_v$ and $b = f_{\text{large}}^{-1}(y_v) \in U_{\text{large}}$ for some $v \in R$. Then, $w_T(a, b) = w_G(\hat{S}_v)$. By Lemma C.1, $\hat{S}_v$ is an approximate Min-$v$-$s$-Cut and $w_T(a, b) \leq \lambda_G(v, s) + \Delta$. To connect this value to $\lambda_G(a, b)$, note that by considering the choices of where $v$ and $s$ lie on the Min-$a$-$b$-Cut, we can show

$$\lambda_G(a, b) \geq \min\{\lambda_G(a, v), \lambda_G(v, s), \lambda_G(s, b)\}. \tag{7}$$

Let $S_a'$ be the $a$ side of the $a$-$v$ cut induced by the approximate Gomory-Hu Steiner tree $(T_v, f_v)$. As $a = f_v^{-1}(x_v)$ and $s \in x_v$, $S_a'$ is also the $s$ side of a $v$-$s$ cut. Therefore, $w_G(S_a') \geq \lambda_G(v, s)$. On the other hand, by our inductive hypothesis and Lemma D.2, this is an approximate Min-$a$-$v$-Cut: $w_G(S_a') \leq \lambda_{G_v}(a, v) + (2i - 2)\Delta \leq \lambda_G(a, v) + (2i - 1)\Delta$. Hence, $\lambda_G(v, s) \leq \lambda_G(a, v) + (2i - 1)\Delta$. The analogous argument holds to show $\lambda_G(v, s) \leq \lambda_G(s, b) + (2i - 1)\Delta$. Therefore,

$$\lambda_G(v, s) \leq \min\{\lambda_G(a, v), \lambda_G(s, b)\} + (2i - 1)\Delta,$$

which further implies $\lambda_G(v, s) \leq \lambda_G(a, b) + (2i - 1)\Delta$ by combining with Equation (7). Therefore,

$$w_T(a, b) \leq \lambda_G(v, s) + \Delta \leq \lambda_G(a, b) + 2i\Delta.$$

In all cases, $w_T(a, b) \leq \lambda_G(a, b) + 2i\Delta$. As the cut corresponding to the edge $(a, b)$ is on the path from $p$ to $q$, it is also a $(p, q)$-cut, so $\lambda_G(p, q) \leq w_T(a, b)$. Furthermore, it must the case that there is an edge $(a', b')$ along the path between $p$ to $q$ such that $a'$ and $b'$ are in different sides of the true Min-$p$-$q$-Cut. Therefore, $\lambda_G(p, q) \geq \lambda_G(a', b')$. As we chose $(a, b)$ to be the minimum weight edge,

$$w_T(a, b) \leq w_T(a', b') \leq \lambda_G(a', b') + 2i\Delta \leq \lambda_G(p, q) + 2i\Delta.$$

This completes the induction. It follows that the call to PrivateGHSteinerTree$(G, V, \varepsilon, 0)$ outputs a $2t_{\max}\Delta$-approximate Gomory-Hu Steiner tree $T$. Substituting in the values $t_{\max} = O(\lg^2 n)$ and $\Delta = O(\frac{n \lg^6 n}{\varepsilon})$ gives the approximation guarantee. $\qquad\square$

We now state our main result on the approximation guarantee of Algorithm 3.

**Theorem D.1.** *Let $T = (V_T, E_T, w_T)$ be the weighted tree output by PrivateGHTree$(G = (V, E, w), \varepsilon)$ on a weighted graph $G$. For each edge $e \in E_T$, define $S_e$ to be the set of vertices of one of the connected components of $T \setminus \{e\}$. Let $u, v \in V$ be distinct vertices and let $e_{\min}$ be an edge on the unique $u$-$v$ path in $T$ such that $w_T(e_{\min})$ is minimal. With high probability, $S_{e_{\min}}$ is an $O(\frac{n \lg^8 n}{\varepsilon})$-approximate Min-$u$-$v$-Cut and moreover, $|\lambda_G(u, v) - w_T(e_{\min})| = O(\frac{n \lg^8 n}{\varepsilon})$.*

*Proof.* Note that for each edge $e$, the final tree weight $w_T(e)$ is obtained by adding noise $\text{Lap}\left(\frac{2(n-1)}{\varepsilon}\right)$ to the cut value $w(S_e)$. Thus, $|w(S_e) - w_T(e)| = O(\frac{n \lg n}{\varepsilon})$ with high probability for all $e \in T$. Now let $e_0$ be an edge on the unique $u$-$v$ path in $T$ such that $w(S_{e_0})$ is minimal. Then, by Lemma D.1, $w(S_{e_0}) \leq \lambda_G(u, v) + O(\frac{n \lg^8 n}{\varepsilon})$ with high probability. As $e_{\min}$ was chosen as an edge on the $u$-$v$ path in $T$ of minimal weight, $w_T(e_{\min}) \leq w_T(e_0)$, and so

$$w(S_{e_{\min}}) \leq w_T(e_{\min}) + O\left(\frac{n \lg n}{\varepsilon}\right) \leq w_T(e_0) + O\left(\frac{n \lg n}{\varepsilon}\right)$$

$$\leq w(S_{e_0}) + O\left(\frac{n \lg n}{\varepsilon}\right) \leq \lambda_G(u, v) + O\left(\frac{n \lg^8 n}{\varepsilon}\right).$$

On the the other hand, $S_{e_{\min}}$ defines a $(u,v)$-cut, so $\lambda_G(u,v) \leq w(S_{e_{\min}})$. This proves the first statement. Moreover, the string of inequalities above combined with $\lambda_G(u,v) \leq w(S_{e_{\min}})$ in particular entails that

$$\lambda_G(u,v) \leq w_T(e_{\min}) + O\left(\frac{n \lg n}{\varepsilon}\right) \leq \lambda_G(u,v) + O\left(\frac{n \lg^8 n}{\varepsilon}\right),$$

from which the final statement follows. $\qquad\square$

## D.2 Privacy via Bounded-Overlap Branching Composition

Consider a family of computation trees parameterized by two differentially private mechanisms, denoted $\mathcal{M}_{\text{recurse}}$ and $\mathcal{M}_{\text{sanitize}}$. Through $\mathcal{M}_{\text{recurse}}$, each node in the tree privately produces both (1) the output of its computation and (2) the topology and description of its children's input. Each child's input consists of a combination of "sensitive" data, transmitted in plaintext, and "sanitized" data, provided as the output of the private mechanism $\mathcal{M}_{\text{sanitize}}$. Let $\sigma(u)$ denote the set of children of a node $u$ in a tree.

Specifically, upon receiving a sensitive dataset $X^u = X_1^u, \ldots, X_s^u$ and a sanitized dataset $Y^u = Y_1^u, \ldots, Y_t^u$ from its parent, a node $u$ computes the following objects via $\mathcal{M}_{\text{recurse}}(X^u, Y^u)$ (we only require that $\mathcal{M}_{\text{recurse}}$ is DP with respect to its first input):

- A number of children $d^u$
- For each child $v \in \sigma(u)$, a set of "sensitive" indices $I^v = \{i_1^v, \ldots, i_n^v\}$
- For each child $v \in \sigma(u)$, a set of "sanitized" indices $J^v = \{j_1^v, \ldots, j_m^v\}$
- An auxiliary output $a^u$

Let $v$ be one of the $d^u$ children of $u$. The input for $v$ consists of a sensitive dataset $X^v = X_{i_1^v}^u, \ldots X_{i_n^v}^u$ and a sanitized dataset, formed by concatenating any function of $Y^u$ with $\mathcal{M}_{\text{sanitize}}(X_{j_1^v}^u, \ldots, X_{j_m^v}^u)$. This setup is illustrated in Figure 2.

Additionally, we consider a privacy model where the indices of the data are public, and two neighboring data sets differ in at most one index. The theorem below holds regardless of how we define the neighborhood relation on a single index. For example, we could say that $X$ and $X'$ are neighboring if there exists at most one index $i$ such that $X_i \neq X_i'$. However, for our main result on DP Gomory-Hu trees, $X = (X_e)_{e \in E} = (w(e))_{e \in E}$ are the weights of edges of the input graph, and two data sets $X, X'$ are neighboring if there exists an $e \in E$ such that for all $e' \in E \setminus \{e\}$, $X_{e'} = X_{e'}'$ and moreover, $|X_e - X_e'| \leq 1$.

**Theorem D.2** (Bounded-Overlap Branching Composition). *Let $\mathcal{M}_{\text{recurse}}$ and $\mathcal{M}_{\text{sanitize}}$ be mechanisms as described, satisfying $(\varepsilon_1, \delta_1)$-DP and $(\varepsilon_2, \delta_2)$-DP, respectively. Let the subsets of indices $\{I^v\}$ and $\{J^v\}$ produced by $\mathcal{M}_{\text{recurse}}$ satisfy the following conditions deterministically for all inputs:*

- *The index sets $\{I^v\}$ are disjoint.*

- *For all indices $j$, $|\{v : j \in J^v\}| \leq \ell$ for a fixed constant $\ell$.*

*Consider the following mechanism $\mathcal{M}_{\text{branch}}$ that, as input, receives a sensitive dataset $X$ and a maximum depth parameter $h$. $\mathcal{M}_{\text{branch}}$ creates a tree $T = (V, E)$ recursively using $\mathcal{M}_{\text{recurse}}$ and $\mathcal{M}_{\text{sanitize}}$ with $X$ as the sensitive input to the root node; the root node does not receive any sanitized input. If the depth of $T$ is greater than or equal to $h$, the mechanism outputs $\perp$. Otherwise, the mechanism releases $T$ along with all outputs $\{(Y^u, d^u, \{I^v\}_{v \in \sigma(u)}, \{J^v\}_{v \in \sigma(u)}, a^u) : u \in V\}$. This mechanism is $(h\varepsilon_1 + (h-1)\ell\varepsilon_2,\ h\delta_1 + (h-1)\ell\delta_2)$-DP.*

*Proof.* Let $N = |X|$. For any $t \in \mathbb{N}$, let $\mathcal{M}_{\text{branch}}^t$ be the mechanism which, given a sensitive dataset $X$, creates a tree by recursively using $\mathcal{M}_{\text{recurse}}$ and $\mathcal{M}_{\text{sanitize}}$ with $X$ as the input to the root node, stopping after recursion depth $t$; we adopt the convention that a tree with a single node has depth $0$. The output of $\mathcal{M}_{\text{branch}}^t$ is the tree $T^t = (V^t, E^t)$ with maximum depth $t$, and the sanitized inputs and node outputs $\{(Y^u, d^u, \{I^v\}_{v \in \sigma(u)}, \{J^v\}_{v \in \sigma(u)}, a^u) : u \in V^t\}$.

Let $\varepsilon(t), \delta(t)$ be the privacy parameters for $\mathcal{M}_{\text{branch}}^t$. The mechanism described in the theorem statement is $(\varepsilon(h-1), \delta(h-1))$-DP as checking whether the tree (with unbounded recursive depth) has

depth at most $h$ can be verified by running the recursion for up to $h$ steps and checking whether $d^u = 0$ for every node $u$ at depth $h$. Outputting either $\perp$ or $(T, \{(Y^u, d^u, \{I^v\}_{v \in \sigma(u)}, \{J^v\}_{v \in \sigma(u)}, a^u) : u \in V^t\})$ is a post-processing of the output of $\mathcal{M}_{\text{branch}}^{h-1}$ and therefore cannot increase the privacy loss.

We prove by induction over $t$ that the following conditions hold:

(a) $\varepsilon(t) \le (t+1)\varepsilon_1 + t\ell\varepsilon_2$

(b) $\delta(t) \le (t+1)\delta_1 + t\ell\delta_2$

(c) Let $S^t \subseteq V$ be the subset of vertices in the tree produced by $\mathcal{M}_{\text{branch}}^t(X)$ at depth $t$. For any $i \in [N]$, there is at most one vertex $u \in S^t$ for which $X_i \in X^u$.

The theorem statement immediately follows from conditions (a) and (b) by plugging in $t = h - 1$.

For the base case, consider $t = 0$. $\mathcal{M}_{\text{branch}}^t(X)$ releases a tree $T^0 = (V^0, E^0)$ as well as $\{(Y^u, d^u, \{I^v\}_{v \in \sigma(u)}, \{J^v\}_{v \in \sigma(u)}, a^u) : u \in V^0\}$. The tree $T^0$ simply contains a single node independently of $X$. Let $u$ be the single node in $V^0$. The root node receives no sanitized dataset, so $Y^u = \emptyset$ independently of $X$. The output $(d^u, \{I^v\}_{v \in \sigma(u)}, \{J^v\}_{v \in \sigma(u)}, a^u)$ is produced by $\mathcal{M}_{\text{recurse}}$ which is $(\varepsilon_1, \delta_1)$-DP. Therefore, the entire mechanism $\mathcal{M}_{\text{branch}}^1$ is $(\varepsilon_1, \delta_1)$-DP. Furthermore, condition (c) of the inductive hypothesis is trivially satisfied as there is a unique vertex.

Consider any $t > 0$. By the inductive hypothesis, releasing the tree $T^{t-1} = (V^{t-1}, E^{t-1})$ along with $\{(Y^u, d^u, \{I^v\}_{v \in \sigma(u)}, \{J^v\}_{v \in \sigma(u)}, a^u) : u \in V^{t-1}\}$ is $(t\varepsilon_1 + (t-1)\ell\varepsilon_2, t\delta_1 + (t-1)\ell\delta_2)$-DP. We will assume that these objects have been released and analyze the privacy of releasing the additional objects output by $\mathcal{M}_{\text{branch}}^t(X)$. Note that the topology of the tree $T^t = (V^t, E^t)$ can already be calculated as post-processing from these objects as the structure of the $t$-th level of the tree is contained in the degrees of the leaves of $T^{t-1}$. Let $S^t = V^t \setminus V^{t-1}$. The additional objects output by $\mathcal{M}_{\text{branch}}^t(X)$ that cannot be obtained via post-processing of $\mathcal{M}_{\text{branch}}^{t-1}(X)$ are $\{(Y^v, d^v, \{I^w\}_{w \in \sigma(v)}, \{J^w\}_{w \in \sigma(v)}, a^v) : v \in S^t\}$.

We will continue by bounding the privacy of releasing these outputs for any node at depth $t$ with respect to the sensitive inputs to the node and its parent. Consider any $v \in S^t$ and let $p(v) \in S^{t-1}$ be the parent of $v$ in $T^t$. The only part of $Y^v$ which cannot be obtained by post-processing $Y^{p(v)}$ is the output of $\mathcal{M}_{\text{sanitize}}(X_{j_1^v}^{p(v)}, \ldots, X_{j_m^v}^{p(v)})$. So releasing $Y^v$ is $(\varepsilon_2, \delta_2)$-DP with respect to the subset of its parent's sensitive input $(X_{j_1^v}^{p(v)}, \ldots, X_{j_m^v}^{p(v)})$ and $(0, 0)$-DP with respect to $X \setminus (X_{j_1^v}^{p(v)}, \ldots, X_{j_m^v}^{p(v)})$. Given the release of $Y^v$ as well as $I^v$ and $J^v$, releasing $(d^v, \{I^w\}_{w \in \sigma(v)}, \{J^w\}_{w \in \sigma(v)}, a^v)$, the output of $\mathcal{M}_{\text{recurse}}(X^v = (X_{i_1^v}^{p(v)}, \ldots, X_{i_n^v}^{p(v)}), Y^v)$, is $(\varepsilon_1, \delta_1)$-DP with respect to $X^v$ and $(0, 0)$-DP with respect to $X \setminus X^v$.

Consider any particular index $i^* \in [N]$ corresponding to the sole index on which two neighboring datasets $X$ and $X'$ differ. By the inductive hypothesis, there is at most one node $u^* \in S^{t-1}$ such that $X_{i^*} \in X^u$. Consider any node $v \in S^t$ which is *not* a child of $u^*$, $p(v) \ne u^*$. Note that $X_{i^*} \notin X^{p(v)}$ implies that $X_{i^*} \notin X^v$ as $X^v \subseteq X^{p(v)}$. By the argument in the paragraph above, conditioning on the release of $\mathcal{M}_{\text{branch}}^{t-1}(X)$, the release of $(Y^v, d^v, \{I^w\}_{w \in \sigma(v)}, \{J^w\}_{w \in \sigma(v)}, a^v)$ does not depend on $X_{i^*}$ and does not affect privacy.

Consider nodes $v \in \sigma(u^*)$, the children of $u^*$. If $i^* \notin I^v$ and $i^* \notin J^v$, as above, conditioning on the release of $\mathcal{M}_{\text{branch}}^{t-1}(X)$, the release of $(Y^v, d^v, \{I^w\}_{w \in \sigma(v)}, \{J^w\}_{w \in \sigma(v)}, a^v)$ does not depend on $X_{i^*}$ and does not affect privacy. If $i^* \in J^v$, the release of $(Y^v, d^v, \{I^w\}_{w \in \sigma(v)}, \{J^w\}_{w \in \sigma(v)}, a^v)$ is $(\varepsilon_2, \delta_2)$-DP. If $i^* \in I^v$, the release of $(Y^v, d^v, \{I^w\}_{w \in \sigma(v)}, \{J^w\}_{w \in \sigma(v)}, a^v)$ is $(\varepsilon_1, \delta_1)$-DP.

By the condition on $\mathcal{M}_{\text{recurse}}$ in the theorem statement, there is at most one child $v \in \sigma(u^*)$ such that $i^* \in I^v$ and at most $\ell$ $v \in \sigma(u^*)$ such that $i^* \in J^v$. This implies that there is at most one node $v \in S^t$ where $i^* \in I^v$, satisfying condition (c) of the inductive hypothesis. As this holds for any choice of $i^*$, conditioned on the release of $\mathcal{M}_{\text{branch}}^{t-1}(X)$, releasing $\{(Y^v, d^v, \{I^w\}_{w \in \sigma(v)}, \{J^w\}_{w \in \sigma(v)}, a^v) : v \in S^t\}$ is $(\varepsilon_1 + \ell\varepsilon_2, \delta_1 + \ell\delta_2)$-DP. By basic composition, summing over the privacy parameters of releasing $\mathcal{M}_{\text{branch}}^{t-1}$, $\varepsilon(t) \le (\varepsilon_1 + \ell\varepsilon_2) + (t\varepsilon_1 + (t-1)\ell\varepsilon_2) = (t+1)\varepsilon_1 + t\ell\varepsilon_2$ and, likewise, $\delta(t) \le (t+1)\delta_1 + t\ell\delta_2$, satisfying conditions (a) and (b) of the inductive hypothesis. This completes the proof. $\qquad\square$

We will use bounded-overlap branching composition to prove the privacy of our GH-tree algorithm.

**Theorem D.3.** *PrivateGHTree$(G, \varepsilon)$, i.e., [Algorithm 3](#), is $\varepsilon$-DP.*

*Proof.* We will first argue that releasing the unweighted tree returned by the call to PrivateGHSteinerTree$(G, V, \varepsilon, 0, n)$ is $(\varepsilon/2)$-DP. To invoke [Theorem D.2](#), we will describe the mechanisms $\mathcal{M}_{\text{recurse}}$ and $\mathcal{M}_{\text{sanitize}}$ for which the corresponding $\mathcal{M}_{\text{branch}}$ mechanism simulates the computation done in this call to PrivateGHSteinerTree.

$\mathcal{M}_{\text{recurse}}$ has a sensitive and sanitized input. Its sanitized input is a list of vertices and edges, edge weights for a subset of its edges, as well as $U, \varepsilon, t, n_{\max}$. Its sensitive input is a set of edge weights for the rest of its edges. Let $G$ be the weighted graph which is formed by combining the two inputs. $\mathcal{M}_{\text{recurse}}$ contracts some vertices (if it is itself a $G_{\text{large}}$ subinstance), picks a uniformly random vertex $s$, calculates $t_{\max}$, and runs PrivateGHTreeStep$\left(G, s, U, \frac{\varepsilon}{4t_{\max}}, \frac{1}{n_{\max}^3}\right)$ to obtain $D, R \subseteq U$, and $\{\hat{S}_v\}_{v \in R}$. The output of $\mathcal{M}_{\text{recurse}}$ is the following:

- Recursive degree $|R| + 1$.

- For each $v \in R$, a recursive child is created which gets sensitive indices corresponding to all edges between pairs of vertices in $\hat{S}_v$. In addition, this child gets sanitized indices corresponding to all edges between $\hat{S}_v$ and $V \setminus \hat{S}_v$.

- An additional recursive child (corresponding to the $G_{\text{large}}$ subinstance) is created which gets sensitive indices corresponding to all edges whose endpoints do not both belong to the same set $\hat{S}_v$. This child has no sanitized indices.

- Auxiliary output $R, \{\hat{S}_v\}_{v \in R}$.

By the privacy of PrivateGHTreeStep in [Lemma C.3](#), $\mathcal{M}_{\text{recurse}}$ is $\left(\frac{\varepsilon}{4t_{\max}}\right)$-DP.

$\mathcal{M}_{\text{sanitize}}$ takes as input a set of edge weights for edges between a set $\hat{S}_v$ and its complement $V \setminus \hat{S}_v$. It contracts $V \setminus \hat{S}_v$ to a vertex $x_v$ and outputs the resulting (summed) edge weight from edges between $\hat{S}_v$ and $x_v$ plus Lap$(\frac{8t_{\max}}{\varepsilon})$ noise. By the privacy of the Laplace mechanism [Theorem A.3](#) and as the contraction/sum operation has sensitivity 1, $\mathcal{M}_{\text{sanitize}}$ is $\left(\frac{\varepsilon}{8t_{\max}}\right)$-DP.

The unweighted tree output by PrivateGHSteinerTree$(G, V, \varepsilon/2, 0, n)$ can be calculated via post-processing of the recursive mechanism $\mathcal{M}_{\text{branch}}$ parameterized by $\mathcal{M}_{\text{recurse}}$ and $\mathcal{M}_{\text{sanitize}}$ with maximum depth $h = t_{\max}$. While in [Algorithm 4](#), we invoke the Combine step to construct the GH-tree, this can easily be simulated given the auxiliary outputs $\{\hat{S}_v\}_{v \in R}$ at each recursive node as these sets determine the GH-tree topology.

The final step to bound the overall privacy of outputting the unweighted tree is to ensure that $\mathcal{M}_{\text{recurse}}$ satisfies the bounded-overlap condition. Consider the edge weights which are the sensitive input to a call to $\mathcal{M}_{\text{recurse}}$. Note that these edge weights are partitioned across the recursive children: either the weight corresponds to an edge with both endpoints in a single $\hat{S}_v$, in which case it is sent to the corresponding $G_v$ subinstance, otherwise it is sent to the special $G_{\text{large}}$ subinstance. Furthermore, any edge weight only belongs to the sanitized indices of at most two recursive children. Each $G_v$ subinstance receives all edge weights for edges between $\hat{S}_v$ and $V \setminus \hat{S}_v$ as sanitized indices. Any particular edge has two endpoints and so can only belong to two such sets.

Plugging in $\ell = 2$ to [Theorem D.2](#), we get the following bound on the privacy of releasing the unweighted approximate GH-tree:

$$h\varepsilon_1 + (h-1)\ell\varepsilon_2 = t_{\max}\left(\frac{\varepsilon}{4t_{\max}}\right) + 2(t_{\max} - 1)\left(\frac{\varepsilon}{8t_{\max}}\right) < \frac{\varepsilon}{2}.$$

In the final algorithm PrivateGHTree$(G, \varepsilon)$, the tree weights are set to be the corresponding cut value in $G$ plus Lap$\left(\frac{2(n-1)}{\varepsilon}\right)$ noise. Note that the choice of which cut values to calculate is a

post-processing of the privatized output of the unweighted tree. As any specific cut value can change by at most 1 in neighboring graphs, outputting a single tree edge weight is $\left(\frac{\varepsilon}{2(n-1)}\right)$-DP. Basic composition over all $n-1$ tree edges means that releasing the edge weights given the tree topology is $(\varepsilon/2)$-DP. In total, releasing the weighted tree is $\varepsilon$-DP, as required. $\qquad\square$

### D.3   Runtime

While runtime is not our main focus, as a final note, our algorithm can be implemented to run in near-quadratic time in the number of vertices of the graph. The runtime is inherited directly from prior work of [4], which utilizes the same recursive algorithm introduced in [61]. The overall structure of their main algorithm and subroutines remains in our work with changes of the form (a) altering runtime-independent conditions in **if** statements or (b) adding noise to cut values or edges in the graph. While left unspecified here, computation of single source Min-$s$-$v$-Cuts in Algorithm 2 should be done via the runtime-optimized algorithm of prior work [85] to achieve the best bound. Then, via Theorem 1.3 of [4], Algorithm 3 runs in time $\tilde{O}(n^2)$ (note that if instead of [85], we use the almost-linear time algorithm for single source Min-$s$-$v$-Cuts [23], the running time would be $n^{2+o(1)}$ for dense graphs).

## E   Additional Related Work

Multiway cut is another cut problem that has been studied in the privacy setting. Given $k$ terminals, the multiway cut problem seeks a partitioning of the graph's nodes into $k$ parts such that (1) each part contains exactly one terminal, and (2) the sum of the weights of edges between parts, known as the cut value, is minimized [28]. The multiway cut problem is NP-hard for $k \geq 3$ [28], implying that all non-private polynomial-time algorithms have approximation factors greater than 1. Dalirrooyfard, Mitrović, and Nevmyvaka [29] present an $\varepsilon$-DP algorithm for the multiway cut problem with a multiplicative approximation factor of 2 and an additive error of $O(n \log k/\varepsilon)$. Chandra et al. [18] provide an $\varepsilon$-DP algorithm achieving an additive error of $\tilde{O}(nk/\varepsilon)$ and a multiplicative approximation ratio that matches the best-known non-private algorithm. They show that an additive error of $\Omega(n \log k/\varepsilon)$ is necessary for any $\varepsilon$-private algorithm for multiway cut. An open question is whether there exists an algorithm with an additive error of $O(n \log k/\varepsilon)$ and a multiplicative approximation ratio that matches the best-known non-private algorithm.

Computing cuts on (suitably defined) graphs is also a key primitive in a wide-range of applications beyond the aforementioned graph algorithms. For example in many clustering problems, cuts are explicitly computed, as in spectral [20, 76] or correlation clustering [69, 65], or the objective involves identifying a small cut implicitly, such as in community detection in the stochastic block model and beyond [40, 24, 43, 42]. Cuts are also at the heart of many learning-theory tasks on graph data, such as learning graph partitions [68, 79], (semi)-supervised learning using cuts [12, 7, 87, 86, 52], active learning [11, 45], and transductive learning [55, 48], to name a few examples. We refer to the papers and references therein for details.

## F   Minimum $k$-cut

Lastly, we note an application to the minimum $k$-cut problem. Here, the goal is to partition the vertex set into $k$ pieces and the cost of a partitioning is the total weight of all edges between different pieces in the partition. We wish to find the smallest cost solution. It is known that simply removing the smallest $k-1$ edges of an exact GH tree gives us a solution to the minimum $k$-cut problem with a multiplicative approximation of 2 [80]. Since we compute an approximate GH tree with additive error $\tilde{O}(n/\varepsilon)$, we can obtain a solution to minimum $k$-cut with multiplicative error 2 and additive error $\tilde{O}(nk/\varepsilon)$. Our corollary is the following.

**Corollary F.1.** *Given a weighted graph $G$ with positive edge weights and a privacy parameter $\varepsilon > 0$, there exists an $\varepsilon$-DP algorithm that outputs a solution to the minimum $k$-cut problem on $G$ in $\tilde{O}(n^2)$ time with multiplicative error 2 and additive error $\tilde{O}(nk/\varepsilon)$ with high probability.*

The only prior non-trivial DP algorithm for the minimum $k$-cut problem is given in [18]. Their pure DP algorithm obtains the optimal additive error $\Theta(k \log(n)/\varepsilon)$ but requires the input graph to be

unweighted while also requiring exponential time. They also give a polynomial time algorithm which also has multiplicative error 2 with additive error $\tilde{O}(k^{1.5}/\varepsilon)$ which holds for weighted graphs, but only works in the *approximate* DP setting. Note that there are also trivial algorithms such as finding the minimum $k$-cut from a solution to the All Cuts problem, e.g., those from Table 1, but this strategy only gives additive error $O(kn^{1.5}/\varepsilon)$ for dense graphs.

Thus, to the best of our knowledge, no prior polynomial-time pure DP algorithm can compute the minimum $k$-cut on weighted graphs with near-linear in $n$ error. Determining the limits of efficient and pure DP algorithms for the minimum $k$-cut problem is an interesting open question.

**Proof of Corollary F.1.** We follow the proof of [80] (via the lecture notes in [19]), replacing the exact GH-tree with our approximate version. The algorithm is simple: we cut the edges corresponding to the union of cuts given by the smallest $k - 1$ edges of our approximate GH-tree $T$ of Theorem 1.1. If this produces more than $k$ pieces, arbitrarily add back cut edges until we reach a $k$-cut.

For the analysis, consider the optimal $k$-cut with partitions $V_1, \ldots, V_k$ and let $w(V_1) \leq \ldots \leq w(V_k)$ denote the weight of the edges leaving each partition without loss of generality. Since every edge in the optimum is adjacent to exactly two pieces of the partition, it follows that $\sum_i w(V_i)$ is *twice* the cost of the optimal $k$-cut. We will now demonstrate $k - 1$ different edges in $T$ which have cost at most $\sum_i w(V_i)$, up to additive error $O(k\Delta) = \tilde{O}(nk/\varepsilon)$, where $\Delta = \tilde{O}(n/\varepsilon)$ is the additive error from Theorem 1.1.

As in the proof in [80], contract the vertices in $V_i$ in $T$ for all $i$. This may create parallel edges, but the resulting graph is connected since $T$ was connected to begin with. Make this graph into a spanning tree by removing parallel edges arbitrarily, root this graph at $V_k$, and orient all edges towards $V_k$.

Consider an arbitrary $V_i$ where $i \neq k$. The 'supernode' for $V_i$ has a unique edge leaving it, which corresponds to a cut between some vertex $v \in V_i$ and some vertex $w \notin V_i$. Since $T$ is an approximate-GH tree, the weight of this edge must be upper bounded by $w(V_i)$ (which is also a valid cut separating $v$ and $w$), up to additive error $\Delta$. The proof now follows by summing across $V_i$. $\square$

# G   Discussion on Open Problems

An interesting open question is whether a better additive error can be achieved or a lower bound established in the setting where we only care about *values* of cuts and not the cuts (vertex bipartitions) themselves. To our knowledge, the best-known error bound for this All-Pairs Min-$s$-$t$-Cut Values problem is $O(n/\varepsilon)$ for *approximate* DP, obtained by a trivial algorithm that adds $\text{Lap}(n/\varepsilon)$ noise to each of the $\binom{n}{2}$ true values; this method satisfies privacy through advanced composition and the Laplace mechanism [34]. Note that this algorithm does not leverage the fact that there are, in fact, at most $n - 1$ distinct cut values or any of the structure of the graph. Before our work, the best algorithm for this problem in the *pure* DP setting solved the All Cuts problem, incurring an $O(n^{3/2}/\varepsilon)$ error [47]. Our work yields an improvement to $\tilde{O}(n/\varepsilon)$ error with pure DP. Both of these solutions in the pure DP setting output the cuts and the values, so it seems probable that better error can be achieved. No non-trivial lower bound is known for this problem. The $\Omega(n)$ lower bound of [29] applies to releasing an approximate Min-$s$-$t$-Cut, whereas releasing a single Min-$s$-$t$-Cut *value* can be achieved with an error of $O(1/\varepsilon)$ using the Laplace mechanism. This question parallels the All-Pairs Shortest-Paths Distances problem studied in [81, 39, 22, 13], where sublinear additive error is achievable for releasing the *values* of all $\binom{n}{2}$ shortest-paths. In contrast, the linear error is required to release any shortest path itself. One might wonder about the sensitivity of a single edge on Min-$s$-$t$-Cut values; specifically, whether two neighboring graphs differ in only a small number of Min-$s$-$t$-Cut values. However, as illustrated in Figure 3, the Min-$s$-$t$-Cut value sets of two neighboring graphs can differ by as many as $n - 1$ entries.

An additional open question is whether there exists a polynomial-time $\varepsilon$-DP algorithm for the global Min Cut problem that achieves error below $\tilde{O}(n/\varepsilon)$. As noted in Corollary 1.2, the polynomial time algorithm from [46] is only approximate DP, though it can be made pure DP if allowed exponential runtime. The same question applies to minimum $k$-cut (see Corollary F.1): what are the limits of efficient, i.e., polynomial-time, algorithms that are also pure DP for this problem?

Finally, in the problem of outputting synthetic graphs that preserve all cuts, the work of [37] establishes a lower bound of $\Omega(\sqrt{mn/\varepsilon})$ on the additive error. However, this result applies to algorithms that do not permit any multiplicative approximation. In the same paper, the authors discuss that an additive error of $\tilde{O}(n)$ with a multiplicative error of $1 + \eta$, for any constant $\eta > 0$, can be achieved in exponential time using the existence of linear-size cut-sparsifiers and the Exponential mechanism. Recently, Aamand et al. [1] gave a polynomial time algorithm for this problem with roughly $n^{1.25}$ additive error with constant multiplicative error. It remains an intriguing open question whether a synthetic graph that preserves all cuts within an additive error of roughly $n$ and a constant multiplicative error can be generated in polynomial time.

