# OpenReview forum: "Differentially Private Gomory-Hu Trees"
_NeurIPS.cc/2025/Conference — NeurIPS 2025 poster_

### Official Review · Reviewer_EWfq · 2025-06-30

**Clarity:** 3
**Significance:** 3
**Originality:** 3
**Rating:** 5
**Confidence:** 4

**Summary:**

This paper presents a purely differentially private (DP) construction of approximate Gomory-Hu tree. As corollaries, they get purely differentially private approximation algorithms for all-pairs min-cuts, global min-cuts, and minimum k-cuts, all with nearly linear additive error which also runs in poly time.

At a high level, the authors closely follow the recent line of work on approximate Gomory-Hu tree construction in [LP21]. But along the way they have to make a number of changes to privatize the algorithm.

First of all, the min-isolating cuts alone are relatively straightforward to privatize. Making it useful for the Gomory-Hu tree application requires an extra contracting property. In Appendix B, the authors introduce line 11 (the middle highlighted part) in Algorithm 1 so that a contracting property is approximately preserved after introducing noises. This contracting property is mainly used to bound the recursion depth, similar to the recursive approximate Gomory-Hu tree analysis in [LP21]. The trick in line 11 here is to introduce extra deterministic weights to the undesirable cuts so as to beat the noisy perturbations. The same trick is also used in Appendix D.

Then Appendix C is mainly the DP version of Cut Threshold Step in [LP21]. An interesting modification is line 7 of Algorithm 2, where they try to include more vertices for smaller isolating cuts.

I have reviewed an earlier version of this paper, and I'm happy to report that significant revisions have been made in improving the exposition of the paper. In particular, the technical overview appears to be substantially rewritten, with very well-structured intuitions along the way.

Overall this looks like a nice paper. Their approach seems to follow [LP21] closely but there are a number of non-trivial modifications to be made for privacy both in the algorithms and in the analysis. In particular, they also have to work with recursive subsets that are not necessarily disjoint, but only known to have bounded overlap.
Target audience would include people who work on differential privacy and graph algorithms in general. The paper also does a decent job in explaining the contexts and giving a high level technical overview. As such, I would vote for acceptance.

**Questions:**

You introduced edge weight differential privacy in Appendix A.5, but in Appendix G you tried to draw an analogy between min s-t cuts and all pairs shortest path distances without mentioning that many existing works considered different privacy notions for these two problems. How are these two problems comparable?

Also, in line 1349-1350, you appear to be describing an obstacle in min s-t cuts. Why exactly is this an obstacle? In particular, for all pairs shortest path distances, even under edge weight differential privacy, the value set of the shortest path distances can also change in at least $\Omega(n)$ entries right? If these problems are somewhat comparable, then this shouldn't really be an obstacle?

Regarding your discussion on synthetic graph, perhaps you want to discuss and/or cite a recent work that goes along the direction that you proposed:
Breaking the $n^{1.5}$ Additive Error Barrier for Private and Efficient Graph Sparsification via Private Expander Decomposition by Anders Aamand, Justin Y. Chen, Mina Dalirrooyfard, Slobodan Mitrović, Yuriy Nevmyvaka, Sandeep Silwal, Yinzhan Xu

**Ethical Concerns:**

["NO or VERY MINOR ethics concerns only"]

**Final Justification:**

*The problem considered is of general interests, many real-world problems can be formulated as cut problems
*The technical approach consists of neat observations and non-trivial modifications to recent developments of fast algorithms for graph cuts; considering the versatility of the isolating cut framework in the design of non-private graph algorithms, I expect many nice applications following a successful example of privatizing the isolating cut framework, and the authors also demonstrated some of these applications.
*Their approach follows [LP21] closely but there are a number of non-trivial modifications to be made for privacy both in the algorithms and in the analysis. In particular, they also have to work with recursive subsets that are not necessarily disjoint, but only known to have bounded overlap. Target audience would include people who work on differential privacy and graph algorithms in general.
*The paper also does a decent job in explaining the contexts and giving a high level technical overview.

As such, I would vote for acceptance.

**Limitations:**

yes

**Paper Formatting Concerns:**

None.

**Quality:**

3

**Strengths And Weaknesses:**

Strengths:
* the problem considered is of general interests, many real-world problems can be formulated as cut problems
* the technical approach consists of neat observations and non-trivial modifications to recent developments of fast algorithms for graph cuts; considering the versatility of the isolating cut framework in the design of non-private graph algorithms, I expect many nice applications following a successful example of privatizing the isolating cut framework, and the authors also demonstrated some of these applications.
* the writing is clear and easy to follow
* all the claims come with rigorous proofs

Weaknesses:
* this is clearly a theoretical paper, with constants and/or poly-log hidden in the bounds; Needless to say, figuring out the constants and supplying numerical evaluations would improve the impacts and outreach of the paper,  and I encourage the authors to do so considering the algorithms are not insanely complicated. For this revew though, I tried to look up the reviewer guidelines but did not see that experiments are explicitly stated as mandatory, where the wordings are phrased as theoretical or experimental supports; so I would not hold the lack of experiments strongly against the authors.

---

> ### Author Rebuttal · Authors · 2025-07-30
>
> >You introduced edge weight differential privacy in Appendix A.5, but in Appendix G you tried to draw an analogy between min s-t cuts and all pairs shortest path distances without mentioning that many existing works considered different privacy notions for these two problems. How are these two problems comparable?
>
> These two problems are not directly related (i.e. as far as we are aware, results for one don't imply results for the other). Our goal was to contrast the progress on the private shortest path problem to the APMC problem (please see the next answer).
> Regarding privacy models, most works on the two problems use the same model of edge privacy, where two graphs are defined to be neighboring if they differ in one edge weight by at most one. Works on DP shortest paths also assume the graph *topology* is public (but the edge weights are private). In our setting, we do not need such assumptions.
>
> >Also, in line 1349-1350, you appear to be describing an obstacle in min s-t cuts. Why exactly is this an obstacle? In particular, for all pairs shortest path distances, even under edge weight differential privacy, the value set of the shortest path distances can also change in at least Omega(n) entries right? If these problems are somewhat comparable, then this shouldn't really be an obstacle?
>
> The discussion on lines 1344-1350 is attempting to motivate a different version of our problem where we only ask to release the pairwise min s-t cut *values*, rather than the cuts themselves. In the DP shortest paths problem, there is a separation between these two versions, as sublinear error is possible for the value problem, but linear error is necessary to release an approximate shortest path. Thus, we are motivated to ask if such a separation also exists for min $st$ cuts.
> The lines starting from “One might wonder…” argue that the simple, but natural, algorithm of just computing pairwise min $st$ values and releasing a noisy version can have large errors since changing even one edge can impact $\Omega(n)$ min cut values. Please see lines 178-181 for further discussion.
> We are not arguing that achieving sublinear error for this values problem is impossible, rather we are hopeful that future work could make progress on this interesting open question.

---

> > ### Comment · Reviewer_EWfq · 2025-07-31
> >
> > Thanks for the clarification. I'm happy to keep my ratings and recommendations.
> >
> > Just out of curiosity, do you happen to have any real-world application in mind for releasing the pairwise min s-t cut values, rather than the cuts themselves?

---

> > > ### Author Response · Authors · 2025-08-04
> > >
> > > We are not aware of any practical applications where only releasing the values suffices. Thus, the values question is mostly of theoretical interest, especially if there is a separation between the values version and the approximate cuts version which we study.

---

### Official Review · Reviewer_FS48 · 2025-07-01

**Clarity:** 2
**Significance:** 3
**Originality:** 2
**Rating:** 3
**Confidence:** 5

**Summary:**

The goal of the paper is to output differentially private s-t-min-cuts for all pairs s,t of an input graph. The paper achieves this by outputting a tree that preserves all s-t-min-cuts with high probability (w.h.p.). The privacy model is edge-weight privacy, where two neighboring graphs differ by exact one unit in the weight of one edge.

The private algorithm is built on top of Li's work on constructing a fast non-private Gomory-Hu tree, particularly incorporating privacy into a subroutine called minimum isolating cuts. This subroutine finds the collection of all minimum cuts that separate each of the nodes in the input set from the remaining nodes. Privacy is ensured by replacing an s-t-min-cut query in the minimum isolating cuts algorithm with a private s-t-min-cut as proposed by Dalirrooyfard et al. The algorithm then artificially injects edges with randomly generated weights into the graph before sampling subgraphs for the next steps, which guarantees certain properties regarding the quality of the selected sets.

Finally, the paper bounds the privacy cost incurred by each edge using a composition theorem on the recursion tree. The main contribution is a private synthetic graph structure that can answer all s-t-min-cuts with a similar additive factor (with an extra the logarithmic factor) as that of answering a single s-t-min-cut in prior work.

**Questions:**

The privacy analysis is my major concern for the correctness of the work. Please answer these questions and provide a more explicit discussion of the privacy guarantees, mostly for Algorithm 1 and 4.

- Algorithm 1 invokes the contraction sub-rountine multiple times. By its definition A1, the subroutine uses edges and edges' weights information. The privacy analysis of Algorithm 1, line 801-804, claims that only the min-cut uses the sensitive information, which is straightforward for the input graph $G$, but not directly for the last call. Can the author comments on the correctness of the privacy analysis, especially analyzing the sensitivity of $\mathcal{H}$?
- Similarly to Algorithm 4 and its privacy analyses on graph $G_v$ and $G_{large}$.
- In Algorithm 2, what does "satisfying the conditions on Algorithm 2" mean?
- In the composition theorem, can the authors specify the privacy model (wrt to the sensitive dataset and sanitized data),i.e., the neighborhood definition wrt the two types of data.

**Ethical Concerns:**

["NO or VERY MINOR ethics concerns only"]

**Final Justification:**

I appreciate the discussion, but I remain firm in my recommendation.

The idea and execution of this paper are promising and have the potential to result in very good work. Therefore, I believe the paper would benefit greatly from a careful revision and should not be rushed for this cycle.

The main justification for this recommendation lies in the correctness of the privacy analysis, which is the most critical aspect of a rigorous privacy method such as differential privacy. I have reviewed the maths carefully and found several mistakes in the privacy analysis—these are somewhat tricky given the context. Although the authors have made some corrections in response, I believe the privacy analysis should be thoroughly reviewed with all formal details, which the authors were unable to provide due to limitations imposed by the conference.

I appreciate the positive opinions of other reviewers regarding this work. However, I opt for a weak reject. If the other reviewers can confidently certify the correctness of the privacy analysis, I will not oppose their judgment. Otherwise, I hope the paper will be carefully revised and reviewed again.

**Limitations:**

Yes.

**Paper Formatting Concerns:**

No.

**Quality:**

3

**Strengths And Weaknesses:**

Strengths:

- Addresses an important problem.
- Provides strong theoretical guarantees, with the utility guarantee close to the theoretical lower bounds (with an additional logarithmic factor).
- The analyses of utility and privacy are non-trivial.
- The composition theorem is interesting and of independent interest.

Weaknesses:

- The algorithm is largely built on two previous works: the non-private Gomory-Hu tree algorithm by Li and the private s-t-min-cut query by Dalirrooyfard et al., with modifications to control the quality of some intermediate objects.
- Some technical details, especially for the privacy analysis, are not clear (will question later)

---

> ### Author Rebuttal · Authors · 2025-07-30
>
> >Weaknesses:
> The algorithm is largely built on two previous works: the non-private Gomory-Hu tree algorithm by Li and the private s-t-min-cut query by Dalirrooyfard et al., with modifications to control the quality of some intermediate objects.
>
> These two works are important to our paper, but we would point out that many DP papers follow the general strategy of carefully privatizing a known non-DP algorithm. In our case, beyond finding the right non-DP algorithm as a starting point and identifying the steps needed to make it private, we had to tackle several technical challenges, which we have highlighted in Section 2 and again below.
>
> Two important examples are (1) ensuring the private min isolating cuts found are close to their non-private counterparts in terms of their weighted edge values and are simultaneously not 'too small' in cardinality and (2) ensuring that a single edge does not appear in many recursive calls of our private GH tree subroutine, even if the recursive depth is bounded.
>
> These challenges are not relevant in the non-private setting, but require novel analysis for a DP algorithm. Furthermore, the Bounded-Overlap Branching Composition technique which we develop along the way may be of independent interest. We refer to Section 2.3-2.6 for further discussion.
>
> >Some technical details, especially for the privacy analysis, are not clear (will question later)
>
> See the discussion below.
>
> >Algorithm 1 invokes the contraction sub-rountine multiple times. By its definition A1, the subroutine uses edges and edges' weights information. The privacy analysis of Algorithm 1, line 801-804, claims that only the min-cut uses the sensitive information, which is straightforward for the input graph G, but not directly for the last call. Can the author comments on the correctness of the privacy analysis, especially analyzing the sensitivity of  H?
>
> First, the sets $W_r$ are private since they are obtained from postprocessing the $\lg|R|$  privately computed min cuts $C_i$. Furthermore, the sets $W_r$ importantly form a partition of $V$. This implies that an edge in the initial graph contributes its weight to at most two edges in $\mathcal{H}$. Namely, an edge internal to some $W_r$ appears only in $H_r$ and an edge between some $W_r$ and $W_{r’}$ is only contracted into an edge in $H_r$ and an edge in $H_{r’}$. Thus, the sensitivity of $\mathcal{H}$ is 2, and running an $\epsilon$-DP algorithm on $\mathcal{H}$ is thus $2\epsilon$-DP with respect to the initial graph. It appears we have forgotten to incorporate this factor of 2 in our privacy analysis. Fortunately, it only affects our bounds by a constant factor, but we apologize for the confusion and will update the manuscript to address this. Thanks for pointing this out!
>
> >Similarly to Algorithm 4 and its privacy analyses on graph Gv and $G_{large}$.
>
> The privacy analysis of Algorithm 4 is more complicated and requires our Bounded-Overlap Branching Composition Theorem D.2 to analyze the privacy loss through all recursive steps of the algorithm. We note (similar to the previous comment) that since the sets $\hat S_v$ are disjoint, any edge of the initial graph can contribute its weight to at most three of the sub-instances $G_v, v\in R$ and $G_{large}$. Moreover, in the language of Theorem D.2, it appears unsanitized in *at most one* of these subinstances, which turns out to be crucial to bound the privacy loss over all recursive steps of the algorithm. Bounding the number of times (sanitized and sensitive) edges appear in recursive sub-instances makes it possible to apply Theorem D.2 to our problem.
>
> >In Algorithm 2, what does "satisfying the conditions on Algorithm 2" mean?
>
> We apologize for the typo. It should say “satisfying the conditions on Line 7 of Algorithm 2”, which is the following condition (stated in the description of Algorithm 2): $\hat{w}(\hat{S}{\_v}^i) \le \hat{\lambda}(s,v)+ (2 \lfloor \log_2 |U| \rfloor - i) + 1) \Gamma_{\text{iso}} + \Gamma_{\text{values}}$. We will update the manuscript to include algorithm lines and fix this typo.
>
> >In the composition theorem, can the authors specify the privacy model (wrt to the sensitive dataset and sanitized data),i.e., the neighborhood definition wrt the two types of data.
>
> The theorem works generally for any replacement definition of neighboring datasets where the indices of the dataset (e.g., the $\binom{n}{2}$ edge weights) are public, but any given datapoint may be replaced with a different value. The key properties we require are that (a) the index set is public and (b) any neighboring datasets $X, X’$ differ at a single index. We will update the manuscript to clarify this.

---

> > ### Comment · Reviewer_FS48 · 2025-08-06
> >
> > Thank you for your rebuttal

---

### Official Review · Reviewer_SqTp · 2025-07-03

**Clarity:** 4
**Significance:** 4
**Originality:** 4
**Rating:** 5
**Confidence:** 4

**Summary:**

Gomory-Hu tree (GH tree) is a data structure that preserves all pairs $s-t$ min cuts of a graph. This paper designs an $\epsilon$-DP algorithm to compute an approximate GH tree with $\tilde{O}(n/\epsilon)$ additive error on each query. Previously, the best $\epsilon$-DP algorithm for all pairs $s-t$ min cuts required an additive error of $O(n^{\frac{3}{2}}/\epsilon)$. Moreover, an existing lower bound shows that even for a single $s-t$ min cut, $\epsilon$-DP algorithm requires $\Omega(n)$ additive error.

**Questions:**

In Line 47, neighbouring weighted graphs are defined as those that have total weights that differ by at most one and in a single edge. I am curious about the number one here. Why can't we scale the weight to make the difference arbitrarily large? Are there other reasonable neighbouring models for weighted graphs?

**Ethical Concerns:**

["NO or VERY MINOR ethics concerns only"]

**Final Justification:**

I have read the rebuttal. I think it addressed most of my concerns, and I will keep the positive rating.

**Limitations:**

Yes

**Quality:**

4

**Strengths And Weaknesses:**

Strengths:
  1. New and improved differentially private algorithm for a fundamental graph problem. Partially match the lower bound.
  2. The paper is written well. I like the first half of section 2, which introduces the existing approaches and the challenges. It is clear to people who are not working in this direction.

Weaknesses:
  1. Pure theory paper, may not be interesting to a large body of the NeurIPS community. Maybe it is helpful to use one or two sentences to discuss the applications of GH tree in machine learning.

---

> ### Author Rebuttal · Authors · 2025-07-30
>
> > Pure theory paper, may not be interesting to a large body of the NeurIPS community. Maybe it is helpful to use one or two sentences to discuss the applications of GH tree in machine learning.
>
> Thank you for the feedback. Indeed, the main focus of our work is theoretical and gives nearly optimal private algorithms for releasing all pairs min $st$ cuts (with error comparable to the lower bound even for releasing a **single** cut).
> That said, there are applied works that use the computation of many min $st$ cuts, and thus their private version might benefit from our work (see, e.g., [1] and [2]). To give a specific example that is relevant to privacy, consider a social network, where each node represents a person and a weighted edge represents the strength of the relationship between two people. Edge-neighboring privacy corresponds to the case where the set of people are known, but we do not want to leak information about any specific friendship relation. Running multiple min $st$ cuts can help in community detection by detecting tightly-knit communities within a network by identifying weak links that separate different groups (see [2] for a non-private version of this example).
>
> [1] Gary William Flake, Robert Endre Tarjan, and Kostas Tsioutsiouliklis. Graph clustering and minimum cut trees. Internet Math., 1(4):385–408, 2003.
>
> [2] Hyungsik Shin, Jeryang Park, and Dongwoo Kang. A graph-cut-based approach to community detection in networks. Applied Sciences, 12(12), 2022.
>
>
> >In Line 47, neighbouring weighted graphs are defined as those that have total weights that differ by at most one and in a single edge. I am curious about the number one here. Why can't we scale the weight to make the difference arbitrarily large? Are there other reasonable neighbouring models for weighted graphs?
>
> Thank you for the feedback. We discuss the choice of neighboring graphs in section A.5.
>
> This is a great point to scale the weight and one can indeed define neighboring graphs to have total weight difference of say $\Delta$, which would result in the additive error also being scaled by $\Delta$. Thus, the “one” is a natural choice to use. This notion of DP is a standard generalization of unweighted edge-DP and has been widely used to study cut problems in graphs (see [1-5] for a small selection of such works).
> Other reasonable DP models include changing an edge (of unbounded weight) or changing the entire neighborhood of a node. But these notions of privacy are too stringent and result in trivial solutions for cut problems. If the edge weights are arbitrary, as in our setting, removing a node or an edge can cause a cut value to go from any positive value to 0. This is a main reason why the literature on preserving cuts uses the edge weight definition, but it is an interesting future direction to explore other natural neighboring notions for weighted graphs.
>
> [1] Anupam Gupta, Katrina Ligett, Frank McSherry, Aaron Roth, and Kunal Talwar. Differentially private combinatorial optimization. In SODA, 2010.
>
> [2] Jeremiah Blocki, Avrim Blum, Anupam Datta, and Or Sheffet. The Johnson-Lindenstrauss transform itself preserves differential privacy. In FOCS, pages 410–419, 2012.
>
> [3] Marek Eliáš, Michael Kapralov, Janardhan Kulkarni, and Yin Tat Lee. Differentially private release of synthetic graphs. In SODA, pages 560–578, 2020.
>
> [4] Mina Dalirrooyfard, Slobodan Mitrović, and Yuriy Nevmyvaka. Nearly tight bounds for differentially private multiway cut. In NeurIPS, 2023.
>
> [5] Jingcheng Liu, Jalaj Upadhyay, and Zongrui Zou. Optimal bounds on private graph approximation. In SODA, pages 1019–1049, 2024.

---

> > ### Comment · Reviewer_SqTp · 2025-08-05
> >
> > I have read the rebuttal. I think it addressed most of my concerns, and I will keep the positive rating.

---

### Decision · Program_Chairs · 2025-09-17

**Decision:**

Accept (poster)

**Comment:**

This paper designs an
edge-differentially private algorithm to compute all s-t minimum cuts in a weighted graph with additive error $\tilde O(n/\varepsilon)$. The adjacency model allows the weight of a single edge to increase or decrease by 1.

Overall, the reviewers were impressed with the result and its presentation. Cuts and related properties are fundamental to graph analysis, and so a relatively clean characterization such as this one feels like a fundamental result.

The reviewers did have several concerns about the current presentation, which I hope the authors will address.

*Correctness.* One reviewer has concerns about correctness, detailed in their review. While the other reviewers were confident that the final result is correct, we encourage the authors to use the comments and discussion to further improve and clarify the presentation.

*Adjacency notion and semantics.* Another major concern was the meaningfulness of the adjacency model for privacy. The authors write, in one response,

> consider a social network, where each node represents a person and a weighted edge represents the strength of the relationship between two people. Edge-neighboring privacy corresponds to the case where the set of people are known, but we do not want to leak information about any specific friendship relation.

The example is not convincing. For the additive error bounds in the paper to be meaningful, the magnitude of the allowed change to one edge's weight must be at most an $\varepsilon$ fraction of the weight of a typical edge. That means only a tiny relative change in the strength of a relationship can be hidden.

A more meaningful example might come from settings where each edge represents a type of person (say, those who work at one endpoint's location and live at the other's), and the weight represents the number of people of that type.

I would encourage the authors to discuss the limitations of this modeling more transparently in the paper. The authors explain their model clearly in the submission's introduction, and discuss it in Appendix A.5. While the discussion elucidates the mathematical model clearly, the only justification for its interest is the fact that it is the most general standard model that allows any approximation at all. Giving an example of data sets where minimum cuts are a useful tool _and_ the adjacency model makes sense is important. If the authors are not aware of any such settings, they should say so.